# Protein aggregates nucleate ice: the example of apoferritin

María Cascajo-Castresana[1,2,3], Robert O. David[3,4], Maiara A. Iriarte-Alonso[2], Alexander M. Bittner[2,5], Claudia Marcolli[3]

[1]División de Salud, TECNALIA, Parque Tecnológico, Paseo de Mikeletegi, 2, 20009 Donostia, Spain
[2]CIC nanoGUNE, Tolosa Hiribidea, 76, 20018 Donostia, Spain
[3]Department of Environmental System Sciences, Institute for Atmospheric and Climate Science, ETH Zurich, 8092 Zurich, Switzerland
[4]Department of Geosciences, University of Oslo, Oslo, 0315, Norway
[5]Ikerbasque, Basque Foundation for Science, Ma Díaz de Haro 3, 48013 Bilbao, Spain

*Correspondence to*: Claudia Marcolli (claudia.marcolli@env.ethz.ch)

**Abstract**

Biological material has gained increasing attention recently as a source of ice-nucleating particles that may account for cloud glaciation at moderate supercooling. While the ice-nucleation (IN) ability of some bacteria can be related to membrane-bound
proteins with epitaxial fit to ice, little is known about the IN active entities present in biological material in general. To elucidate the potential of proteins and viruses to contribute to the IN activity of biological material, we performed bulk freezing experiments with the newly developed drop freezing assay DRINCZ, which allows the simultaneous cooling of 96 sample aliquots in a chilled ethanol bath. We performed a screening of common proteins, namely the iron storage protein ferritin and its iron-free counterpart apoferritin, the milk protein casein, the egg protein ovalbumin, two hydrophobins, and a yeast ice-
binding protein, all of which revealed IN activity with active site densities >0.1 mg$^{-1}$ at -10°C. The tobacco mosaic virus, a plant virus based on helically assembled proteins, also proved to be IN active with active site densities increasing from 100 mg$^{-1}$ at -14°C to 10,000 mg$^{-1}$ at -20°C. Among the screened proteins, the IN activity of horse spleen ferritin and apoferritin, which form cages of 24 co-assembled protein subunits, proved to be outstanding with active site densities >10 mg$^{-1}$ at -5°C. Investigation of the pH dependence and heat resistance of the apoferritin sample confirmed the proteinaceous nature of its IN
active entities but excluded the correctly folded cage monomer as the IN active species. A dilution series of apoferritin in water revealed two distinct freezing ranges, an upper one from -4 to -11°C and a lower one from -11 to -21°C. Dynamic light scattering measurements related the upper freezing range to ice-nucleating sites residing on aggregates and the lower freezing range to sites located on misfolded cage monomers or oligomers. The sites proved to persist during several freeze-thaw cycles performed with the same sample aliquots. Based on these results, IN activity seems to be a common feature of diverse proteins,
irrespective of their function, but arising only rarely, most probably through defective folding or aggregation to structures that are IN active.

## 1 Introduction

The formation and glaciation of mixed-phase clouds influence radiative transfer, and eventually initiate precipitation, thus determining cloud lifetime (DeMott et al., 2010; Mülmenstädt et al., 2015; Matus, and l'Ecuyer, 2017; Kanji et al., 2017).
Clouds may glaciate through the homogeneous freezing of cloud droplets when air masses cool below -36°C, or at higher temperature in the presence of ice-nucleating particles (INPs). The characterization of INPs is a critical step towards understanding and predicting the climatic impacts of clouds (DeMott and Prenni, 2010).
When an INP is immersed in a cloud droplet, freezing occurs on the INP's surface through heterogeneous nucleation when the temperature falls below the threshold value for activation. This immersion freezing mechanism is considered to be the most
common pathway to cloud glaciation (de Boer et al., 2010; Westbrook and Illingworth, 2013). Alternatively, cloud droplets may freeze while they come in contact with an INP (contact freezing) or while an INP activates to a cloud droplet (condensation freezing) (Murray et al., 2012; Vali et al., 2015; Kanji et al., 2017). After the formation of the first ice crystals, cloud glaciation

may proceed through additional primary ice nucleation occurring on INPs that become active at lower temperatures, ice crystal multiplication (Hallett and Mossop, 1974; Yano and Phillips, 2011; Crawford et al., 2012; Lauber et al., 2018; Field et al., 2017), and the Wegener-Bergeron-Findeisen process (Wegener, 1911; Bergeron, 1928; Findeisen, 1938; Korolev, 2007; Korolev and Field, 2008).

INPs are a very small subgroup of atmospheric aerosol particles; they may represent just 1 in a million or even fewer particles of the whole aerosol population (DeMott et al., 2010). The best-established class of INPs are mineral dusts (Murray et al., 2012; Kanji et al., 2017), originating mainly from arid regions comprising the global dust belt which stretches from the Sahara to the Taklimakan (Sassen et al., 2003; Prospero et al., 2002; Ginoux et al., 2012; Engelstaedter et al., 2006). However, most mineral particles exhibit significant ice-nucleation (IN) activity only below −15°C (Hoose and Möhler, 2012; Murray et al.
2012; Aktinson et al., 2013; Kanji et al., 2017). At higher temperatures, most atmospheric INP that have been identified so far are of biological origin (Levin and Yankofsky, 1983; Murray et al., 2012; Kanji et al., 2017).

One source of biogenic INP is soil organic matter containing plant litter, remains of micro-organisms, lipids, carbohydrates, peptides, cellulose, lignin and humic substances (Simoneit et al., 2004; Oades, 1993; Conen et al., 2011; O'Sullivan et al., 2014; Hiranuma et al., 2015a; Rigg et al., 2013; Wang and Knopf, 2011). Soil dusts consisting of mineral particles mixed with
a small fraction of soil organic matter have been shown to nucleate ice at higher temperature than bare mineral dusts (Conen et al., 2011; O'Sullivan et al., 2014; Steinke et al., 2016; Knopf et al., 2018). Dust emanating from agricultural sources has been estimated to contribute around 20 % to the global dust burden (O'Sullivan et al., 2014; Tegen et al., 2004; Zender et al., 2004). Recent studies suggest that perturbations of the soil and plant surfaces lead to the release of biological organisms that can serve as INP (Huffman et al., 2013; Prenni et al., 2013; Tobo et al., 2013; DeMott et al., 2016). The biological origin of
IN activity of soil dusts above -15°C is usually inferred from the decrease in freezing temperature after heat treatment or digestion with hydrogen peroxide (Du et al., 2017; O'Sullivan et al., 2014; Hill et al., 2016).

The sea surface has also been examined as a source for biogenic INP (Wilson et al, 2015; Ladino et al., 2016). Atmospheric INP concentrations measured on ships were found to be influenced by local marine biological activity and sea spray production (Bigg, 1973; Burrows et al., 2013). During a phytoplankton bloom, Wang et al. (2015) observed an increase in IN activity of
sea spray aerosol above -15°C. Marine phytoplankton has been found IN active, both intact cells and exudates (Schnell, 1975; Alpert et al., 2011; Wilson et al., 2015; Ladino et al., 2016).

Biological INPs include fungal spores, pollen, viruses, microorganisms like bacteria, algae, lichens and archaea, and fragments, exudates, and excretions of microorganisms, plants and animals (Murray et al., 2012, Morris et al., 2013a; Després et al., 2012; Kanji et al., 2017). Bacterial IN activity was found in *Pseudomonas* and related species like *Xanthomonadaceae* (Kim et al.,
1987) and Enterobacteriaceae (Lindow et al., 1978) but rarely outside the Gammaproteobacteria (Ponder et al., 2005; Mortazavi et al., 2008; Failor et al., 2017). *Pseudomonas syringae* (*P. syringae*) are the best investigated IN active bacteria. They have been isolated from decaying leaf litter and can induce freezing at temperatures up to -2°C (Schnell and Vali, 1972; Maki et al., 1974; Vali et al., 1976; Möhler et al., 2007). *P. syringae* are gram-negative bacteria that populate leaf surfaces and are able to cause frost injuries in plants (Lindow, 1983; Hirano and Upper, 2000; Akila et al., 2018). They were shown to owe
their IN activity to a protein located on the outer cell membrane that templates ice through a sequence of amino acids providing an epitaxial fit to ice (Kajava and Lindow, 1992; Murray et al., 2012). IN activity is preserved when the cells are disrupted, though with a shift to lower freezing temperatures (Govindarajan and Lindow, 1988). At lower temperatures, also other types of bacteria (including gram-positive ones) proved to exhibit IN activity (Ponder et al., 2005; Mortazavi et al., 2008; Failor et al., 2017; Akila et al., 2018).

Screening experiments revealed IN activity of lichen samples from a variety of locations with freezing onset temperatures up to -5°C (Moffett et al., 2015), and even up to -2.3°C (Kieft, 1988). The IN activity was found to originate primarily from the mycobiont (Kieft and Ahmadjian, 1989), providing evidence for a fungal rather than bacterial source of IN activity (Kieft and Ruscetti, 1990). The sites seem to be proteinaceous, although they are less sensitive to heat and pH variation compared with

the ice nucleating proteins expressed by *P. syringae* (Kieft and Ahmadjian, 1989; Kieft and Ruscetti, 1990; 1992). In screening experiments, most fungi failed to show IN activity above -20°C with few exceptions such as *Fusarium acuminatum* and *Fusarium avenaceum* (Pouleur et al., 1992; Pummer et al., 2013; Haga et al. 2013; 2014). Yet, IN active fungi with freezing onsets as high as -5°C could be identified in bioaerosols (Huffman et al., 2013) and in soils (Fröhlich-Nowoisky et al., 2015).

Heat resistance and insensitivity to pH variation suggests that the IN active entity is more similar to the ones of lichen than to bacterial ones (Pouleur et al., 1992). Surveys of the IN ability of pollen showed that only few types were active, the most active ones stemming from birch and conifer trees, yet, only at temperatures below -9°C (Diehl et al., 2001; von Blohn et al., 2005; Pummer et al., 2012). Intriguingly, water which has been in contact with pollen and then been separated, nucleated ice as efficiently as the whole pollen grains themselves. Moreover, IN activity has also been found in aqueous extracts of birch leaves

and branches (Felgitsch et al., 2018).

Heterogeneous ice nucleation is considered to arise from the ability of surfaces to order water molecules in an ice-like pattern. The arrangement of water molecules at a surface depends on surface charge and functional groups (Glatz and Sarupria, 2016; Abdelmonem et al., 2017; Pummer et al., 2015). A relevant role is attributed to surface OH and NH groups that are able to form hydrogen bonds to water molecules. Their number and arrangement have been used to explain IN activity of different

mineral surfaces (Pedevilla et al., 2007; Hu and Michaelides, 2007; Glatz and Sarupria, 2018; Kumar et al., 2019b). A lattice match between ice and the ice-nucleating agent is often considered a prerequisite for heterogeneous ice nucleation. Yet, while some IN active substances such as AgI (Marcolli et al., 2016) and 2D-crystalline films formed by long-chain alcohols (Popovitz-Biro et al., 1994; Zobrist et al., 2007; Qiu et al., 2017) exhibit a lattice match, others such as quartz (Kumar et al., 2019a) do not, and even others such as $BaF_2$ exhibit a lattice match but fail to be IN active (Conrad et al., 2005). The difficulty

to pinpoint surface properties that are required for heterogeneous ice nucleation may be explained by growing evidence that it is not the whole surface that is able to nucleate ice but just special nucleation sites (Vali, 2014; Vali et al., 2015), which may arise through defects or impurities. Applying classical nucleation theory to heterogeneous ice nucleation yields nucleation site areas in the range of $10 - 50$ nm$^2$ required to host an ice embryo of critical size (Kaufmann et al., 2017).

Taking surfaces that are large enough to host a critical ice embryo and have the ability to form hydrogen bonds to water

molecules as requirements for IN activity, organic molecules with hydroxyl or carboxyl functionalities should potentially be able to induce freezing (Pummer et al., 2015). Indeed, microcrystalline cellulose has been found to nucleate ice up to -9°C (Hiranuma et al., 2015a). The IN activity of birch tree extracts stems from macromolecules or aggregates of macromolecules which involve polysaccharides (Pummer et al., 2012) and proteins (Tong et al., 2015; Felgitsch et al., 2018) that may coaggregate. Similarly, the exudate material acting as INP in marine aerosol (Wilson et al., 2015; Ladino et al., 2016) was

found to contain polysaccharidic and proteinaceous compounds (Aller et al., 2017). Finally, ice-nucleating proteins expressed by *Pseudomonas* exhibit a repetition unit containing threonine amino acids with hydroxyl functional groups that are able to template ice. Aggregates involving only few of these proteins are water soluble and induce ice nucleation up to -7°C. Larger aggregates nucleate ice up to -2°C but require the intact outer cell membrane to be stable (Polen et al., 2016; Zachariassen and Kristiansen, 2000).

So far, investigations have been focused on proteins that are expressed by organisms to nucleate ice. Here we examine whether proteins as a type of macromolecules have an inherent ability to nucleate ice.

To elucidate the potential of proteins and viruses to contribute to the IN activity of biological material, we employed DRINCZ, a newly developed drop freezing assay (David et al., 2019) to screen the IN activity of common proteins, namely the iron storage protein ferritin and its iron-free counterpart apoferritin (with protein subunits assembled to a cage), the milk protein

casein (in solution producing assembled casein micelles), the egg protein ovalbumin, the hydrophobins HPA and HPB, and the ice-binding protein LeIBP, produced by the yeast Leucosporidium. In addition, we also investigated the IN activity of the tobacco mosaic virus (TMV), a common plant virus, present in plants all over the world. Fillhart et al. (1997) showed that the nearly identical tomato mosaic Tobamovirus (ToMV) can be spread by fog (Fillhart et al., 1997). While all of the proteins and

the virus exhibited IN activity, ferritin and apoferritin proved to be outstanding with freezing onsets as high as -4°C. Therefore, we focused in the following on the elucidation of the origin of the IN activity of ferritin and apoferritin samples.

## 2 Material and methods

### 2.1. Description of the proteins and the virus

#### 2.1.1 Ferritin and apoferritin

Ferritin is composed of 24 (protein) subunits, which co-assemble into a protein shell with an inner cavity of about 7 – 8 nm in diameter, hosting up to 4500 iron atoms ($Fe^{3+}$) in form of an amorphous oxide, and an outer diameter of around 12 nm (see Fig. 1 for its structure). In bacteria and plants, ferritin is formed by 24 identical subunits assembled into a 432-point symmetric hollow shell (Aumiller et al., 2018; Ghirlando et al., 2016; Zeth et al., 2016). In mammals, the apoferritin cage is composed of L (light) and H (heavy) subunits, in a tissue-specific stoichiometry. The L-type subunit (M ≈ 20 kDa) is enriched in ferritin isolated from liver and spleen and contains a mineral nucleation site, while the H-type subunit (M ≈ 21 kDa) contains a ferroxidase site and is more numerous in ferritin isolated from heart and skeletal muscles (May et al., 2010). The H- and L-subunits are isomorphous and share the same tertiary structure with a bundle of four antiparallel α-helices, a shorter helix on top of them (see Fig.1), and loops connecting the helices (Stefanini et al., 1996; Massover, 1993). The subunits are roughly cylindrical, a little more than 5 nm long, and 2.5 nm wide. The L-subunits provide the assembled molecule a greater stability towards chemical and physical agents than do the H-subunits (Yoshizawa et al., 2007). Ferritin and apoferritin exhibit channels at the intersection of the subunits, through which certain ions or molecules can travel. These channels are critical for ferritin's ability to release iron in a controlled fashion.

In this study, commercially available ferritin isolated from horse spleen is used. Apoferritin is obtained from ferritin by removing the iron oxide. Horse spleen apoferritin consists of 85 – 90 % L and 10 – 15 % H chains (Stefanini et al., 1996; May et al., 2010). From sedimentation velocity measurements molar masses of 440 – 500 kDa (Thomas et al., 1998; May et al., 2010; Ghirlando et al., 2016) are estimated for apoferritin while the calculation based on the subunit molar masses determined from cDNA sequences (H-type: 21,269 Da, L-type: 19'978 Da) yields 481.2 kDa (assuming 85 % L- and 15 % H-chains).

Two different batches of horse spleen ferritin and apoferritin saline solutions (0.15 M NaCl and 0.135 M NaCl, respectively) were used (0.2 µm filtered). Both were purchased from Sigma-Aldrich (product number A3641 for apoferritin and F4503 for ferritin). Batches 1 of ferritin and apoferritin were used for pH variation and stress experiments. Batch 1 of apoferritin with batch number SLBD5084V and the quality release date of January 10, 2013, has a concentration specified by Sigma-Aldrich of 37 mg/ml, while our own measurements by the Bradford protein assay (Bradford, 1976) yielded 33.2 mg/ml. Batch 1 of ferritin with the batch number SLBQ9541V and a release date of June 30, 2016 has a concentration specified by Sigma-Aldrich of 55 mg/ml compared to our own measurements yielding 49.7 mg/ml. Note that for ferritin we provide the protein mass concentration, the iron oxide is not counted to provide better comparison to the other proteins. Batch 2 of apoferritin (batch number SLBR2614V) was used for the dilution series, the disassembly-reassembly experiment, and the refreeze experiments. It has the release date of August 25, 2016, and a specified concentration of 43 mg/ml. Batch 2 of ferritin (batch number SLBV7127) with a specified concentration of 61 mg/ml has the quality release date of December 13, 2017. The saline solutions purchased from Sigma-Aldrich were diluted with pure water (purchased from Sigma-Aldrich) for IN experiments. Apoferritin solutions are colourless whereas ferritin solutions present a yellow/orange colour due to the presence of $Fe^{3+}$ (see Fig. S1 of Supplementary Information).

A part of batches 2 of ferritin and apoferritin was dialyzed against ammonium bicarbonate buffer for 96 h. For this purpose, the samples were suspended in 10 mM ammonium bicarbonate (Sigma-Aldrich, 09830) pH 7.4 – 7.6 prepared with Milli-Q

water. Dialysis was achieved by using 10000 MWCO dialysis cassettes (Thermo Scientific) for a period of 96 h, with the ammonium bicarbonate buffer replaced every 24 h.

### 2.1.2 Ovalbumin

Ovalbumin is a protein found in large quantities in avian egg white, most probably serving as a biological reserve of amino acids. It is a non-inhibitory member of the serine protease inhibitor (serpin) superfamily with a molecular weight of ~44.3 kDa (385 amino acids) (Huntington and Stein, 2001; Stein et al., 1991). Ovalbumin from chicken egg white was purchased from Sigma-Aldrich (A5503) as a lyophilized powder (≥ 98% purity).

### 2.1.3 Casein

Casein is a major component of milk giving it its white colour and unique texture (e.g. Ozeki et al., 2009). There are four types of caseins, namely $\alpha_{s1}$-, $\alpha_{s2}$-, $\beta$-, and $\kappa$-casein with molecular weights between 19 and 25 kDa. These proteins adopt flexible conformations, albeit with significant amounts of secondary and, probably, tertiary structure (Swaisgood, 1993; Sunde et al., 2017). All four types of caseins together with colloidal calcium phosphate associate to highly hydrated micelles with average diameters of 150 to 200 nm (Dalgleish and Corredig, 2012) as shown in Fig. 1. The association is governed by weak hydrophobic interactions between casein proteins and by binding of calcium through the phosphoserine groups of $\alpha_{s1}$-, $\alpha_{s2}$-, and $\beta$-caseins leading to the formation of calcium phosphate nanoclusters within the micelles (Lucey and Horne, 2018). Phosphorylated serines are lacking in $\kappa$-casein, which is located on the outer surface of the casein micelles (Sunde et al, 2017). Casein from bovine milk containing all types of casein was used in this work (Sigma, C7078; technical grade).

### 2.1.4 Hydrophobins HPA and HPB

Hydrophobins are small cysteine-rich proteins of about 100 amino acids (MW ≈ 10 kDa) that are secreted by filamentous fungi. They can self-assemble into amphipathic monolayers on hydrophobic and hydrophilic surfaces as well as on interfaces (Morris et al., 2013b). Class I and Class II hydrophobins are discriminated based on their hydropathy patterns and stability towards solvents and detergents (Wessels, 1996; Wohlleben et al., 2010). In Fig. 1, the structure of a class I hydrophobin, DewA, is depicted.

In this study, fusion hydrophobins H*Protein A (HPA) and H*Protein B (HPB) supplied by BASF (Ludwigshafen, Germany) were used. These hydrophobins combine the class I hydrophobin DewA of *Aspergillus nidulans* and the synthase yaaD protein of *Bacillus subtilis* as fusion partners. HPA contains the whole yaaD protein, HPB only a truncated form (Wohlleben et al., 2010). Both HPA and HPB carry a hexahistidine terminus.

### 2.1.5 Ice-binding protein LeIBP

LeIBP is a glycosylated ice-binding protein with a molecular mass of ~25 kDa that is produced by Arctic yeast *Leucosporidium* sp. AY30 (Lee et al., 2012). It consists of a right handed $\beta$-helix fold, a long helix ($\alpha$3) and a C-terminal hydrophobic loop. The $\beta$-helical fold features aligned Thr/Ser/Ala residues that are considered critical for ice binding. LeIBP forms dimers in solution, most probably via the hydrophobic surfaces of helix $\alpha$3 and the C-terminal loop, thus concealing the hydrophobic areas from the solvent (Lee et al., 2012) as shown in Fig. 1. LeIBP used in this study was supplied by Dr. Se Jong Han from the Korea Polar Research Institute (KOPRI).

### 2.1.6 Tobacco mosaic virus (TMV)

The Tobacco mosaic virus (TMV) is assembled from a single-stranded RNA (making up only 5% of the mass), enveloped in 2100 identical helically arranged proteins. The thus formed hollow tube is 300 nm in length, with an external diameter of 18

nm (Eleta-Lopez and Calò, 2017; Alonso et al., 2013). TMV infects plants of the family of Solanaceae such us tobacco, tomato or pepper, causing characteristic mosaic-like patterns, it is harmless to mammals. A TMV suspension (10 mg/mL) was provided by Prof. Christina Wege (University of Stuttgart, Germany)

## 2.2 Freezing experiments performed with DRINCZ

Drop freezing assays investigate heterogeneous ice nucleation in an array of droplets of microliter volumes, and are able to detect low concentrations of INP. Droplet freezing experiments were first reported by Vali and Stansbury (1966) and have since then been used in numerous studies (e.g. Stopelli et al., 2014; Hill et al., 2014; Hiranuma et al., 2015b; Budke and Koop, 2015; Tobo, 2016).

The recently developed DRoplet Ice Nuclei Counter Zürich (DRINCZ) is used for IN measurements (David et al., 2019) in

this study. The drop freezing setup consists of four main parts: (i) a 96-well tray containing in each well 50 µL of liquid sample, (ii) a recirculating chiller bath filled with ethanol to cool the sample, (iii) LED lights and an USB camera to observe the freezing of the wells and (iv) a computer to control the sample temperature and cooling rate, as well as to record and evaluate pictures of the freezing wells.

A home-made lamp built out of LED strips enclosed in an ethanol proof housing is submerged in the cooler liquid to illuminate

the 96-well tray from below. The USB camera is placed above the chiller and directed toward the tray. Images are recorded every 15 s, which corresponds to a picture taken every 0.25˚C, when the bath is cooled with 1˚C/min.

In a typical experiment, 50 µl aliquots of the sample solutions are pipetted with an automatic eight-channel pipette into the 96-well tray, consisting of 8 x 12 wells of 200 µl (732-2386, VWR, USA). The wells are sealed with a transparent sealable foil (Platemax® CyclerSeal Sealing film, Axygen Inc.) to prevent any impurities from settling into the samples. The tray is placed

in the ethanol bath of the chiller (LAUDA Proline RP 845 Refrigerating Circulator, Lauda-Königshofen, Germany). A temperature ramp (-1°C/min) is adjusted via the control software (LabVIEW). During the freezing process the wells turn dark, because small ice crystals scatter light more effectively than liquid water. This decrease in transmission is evaluated automatically by a MATLAB code to detect the initial decrease of brightness which is taken as the instant of IN (see David et al., 2019, for a detailed description). For the measurements performed with batch 2 (dilution series, disassembly-reassembly,

and refreeze experiments) the bath leveler, which keeps the ethanol bath level constant during a cooling ramp, was used as described in David et al. (2019). Protein and virus screening and experiments with batches 1 of ferritin and apoferritin were performed without the bath leveler. In order to correct the temperature difference between the samples within the 96-well tray and the temperature reported by the chiller, a temperature correction was performed as described in David et al. (2019).

Frozen fractions (FF) were converted to cumulative active sites as given in Vali (2019):

$$K(T) = \frac{1}{V} \cdot (lnN_0 - lnN(T)) \,, \tag{1}$$

with $N_0$ and N(T) being the total number of wells (96) and the number of frozen wells at temperature T, respectively, and V is the volume of each aliquot. Differential active site densities were calculated as:

$$k(T) = \frac{1}{V \cdot \Delta T} \ln\left(1 - \frac{\Delta N}{N(T)}\right) \tag{2}$$

where ΔN is the number of wells freezing within the temperature interval ΔT. The cumulative active site density is obtained

from the differential one through:

$$K(T) = \sum_0^T k(T) \cdot \Delta T. \tag{3}$$

When FF curves overlapped with freezing of water devoid of sample, a background correction was performed by subtracting the differential active site density k(T) of the background from the one of the sample as outlined in Vali (2019) and David et al. (2019).

## 2.3 Sample preparation

### 2.3.1 Screening experiments

For screening experiments solutions were prepared with Sigma-Aldrich (SA) water (molecular biology reagent water from Sigma-Aldrich).

### 2.3.2 pH variations

Six different buffers with pH values between 0 and 9.5 were prepared (see Table 1). The buffer at pH 0 was prepared adding 8.58 ml HCl (hydrochloric acid 37%, Merck KGaA, Darmstadt) to 100 ml of SA water. The buffer at pH 2 was prepared with KCl (Potassium chloride >99.5%, Sigma-Aldrich, Missouri, USA) and HCl (37 %). The pH 3.5 and pH 5 buffers were prepared using citric acid ($C_6H_8O_7$, 99%, Sigma-Aldrich, Missouri, USA) and $Na_2HPO_4 \cdot 7H_2O$ (sodium phosphate dibasic heptahydrate, Sigma-Aldrich, Missouri, USA). The buffer at pH 7 was prepared with HEPES ($C_8H_{18}N_2O_4S$, >99.5%, Sigma-Aldrich, Missouri, USA) and NaOH (sodium hydroxide, >98%, Sigma Aldrich, Missouri, USA). Buffer pH 9.5 was prepared with $Na_2B_4O_7 \cdot 10H_2O$ (sodium tetraborate decahydrate, Merck KGaA, Darmstadt) and NaOH (>98%). All pH values were verified with a pH meter (691 pH Meter, Metrohm, Swiss).

Two different concentrations of apoferritin (0.34 mg/ml and 0.036 mg/ml) and ferritin (0.39 mg/ml and 0.04 mg/ml protein) solutions were prepared from batches 1 of apoferritin and ferritin. The precise amount of ferritin and apoferritin was added to the various buffers to assess the effect of pH on the IN activity. Samples were kept in these buffers overnight. The pH of each solution was measured before a freezing experiment was carried out. The results are shown in Table 1.

## 2.4 Stress treatments

For the heat treatment, apoferritin solutions (batch 1, 0.34 mg/ml) were prepared with SA water and heated to 110°C for 5 hours. To prevent water loss, the bottles were loosely covered by a cap. For the combined heat and low pH treatment, a solution of 0.34 mg/ml concentration (batch 1) was prepared in pH 0 buffer and submitted to the same heat treatment. We used glass beakers closed with a loosely screwed stopper to prevent overpressure.

## 2.5 Disassembly-reassembly experiments

A two-step solution preparation procedure was used to achieve disassembly and reassembly.

For the disassembly experiment, the apoferritin solution (batch 2) was diluted with pH 2 buffer to prepare pH 2 apoferritin solutions with 0.036 mg/ml and 0.018 mg/ml concentrations. These solutions were allowed to rest for 1 h before filling the 96-well tray for DRINCZ freezing measurements.

For disassembly-reassembly experiments, the apoferritin solution (batch 2) was diluted with the pH 2 buffer to prepare pH 2 apoferritin solutions with 0.072 mg/ml and 0.036 mg/ml concentrations, and subsequently allowed to rest for 1 h. For reassembly, 0.1 M NaOH was added to the pH 2 solutions until reaching pH 8. Then SA water was added to obtain the desired apoferritin concentrations of 0.036 mg/ml and 0.018 mg/ml. Before preparation for the DRINCZ experiments, the solutions were allowed to rest for 30 min.

## 2.6 Refreeze experiments

Apoferritin solutions (batch 2, concentrations of 0.34 mg/ml, 0.036 mg/ml and 0.018 mg/ml) were prepared and tested for IN activity in DRINCZ. The IN activity of the same 96-well tray was tested again during the following four days in refreeze experiments. Between experiments, the tray was stored at 4°C.

**2.7 Dynamic light scattering**

The hydrodynamic diameter of nanostructures in protein solutions was determined by dynamic light scattering (DLS) using a Zetasizer Nano-ZS (Malvern Instruments Ltd., Malvern, Great Britain). Three runs were performed in three replicates, resulting in nine measurements per sample. To obtain consistent results, noise was reduced by increasing measurement times for low concentration samples. The same procedure was followed for reference solutions of polystyrene latex and gold nanoparticles of known diameters (see Fig. S2 in SI). The hydrodynamic diameter (z-average) and volume-weighted distribution (Stetefeld et al., 2016) of protein assemblies were calculated with the equipment software (v.7.12, Malvern Instruments Ltd., Malvern, Great Britain) without any further data processing, hence assuming spherical shapes. Size-resolved concentrations were obtained by multiplying the volume-weighted distribution by the solution concentration.

## 3 Results and discussion

### 3.1 IN activity screening of common proteins and a virus

All investigated proteins (and TMV) induced freezing clearly above the reference curve of pure SA water, given as the grey line in Fig. 2a. However, they exhibited large variations in onset (-4 to -12°C) and complete freezing ($FF = 1$) temperatures (-7°C to -23°C). Apoferritin proved to be the most IN-active sample with a freezing onset of -4°C and complete freezing at -7°C despite being less concentrated (0.34 mg/ml) than the other proteins (1 mg/ml). The milk protein casein showed a similarly steep freezing curve as apoferritin and ferritin, however, shifted to lower temperatures with an onset at -8°C and complete freezing at -13°C. The freezing curves of the egg protein ovalbumin, the ice-binding protein LeIBP, and the hydrophobins HPA and HPB all exhibit freezing onsets between -6°C and -8°C, a plateau at about -10°C, followed by a steeper increase, resulting in $FF = 1$ between -19°C and - 23°C. This indicates the presence of two different types of sites: rare ones with activity above -10°C, and more common ones with activity between -10 and -23°C. The fusion hydrophobin HPB, which contains only a truncated version of the yaaD protein, is more IN-active than HPA, which contains the full yaaD protein, suggesting that the hydrophobin part of the fusion protein is relevant for the observed IN activity. TMV shows a lower IN activity than the proteins with freezing onset only at -12°C, however, it is also the most dilute sample with a concentration of only 0.002 mg/ml. If we compare the ice nucleation activity in terms of cumulative active site densities (Fig. 2b), TMV exhibits a higher active site density than the hydrophobins, the ice-binding protein, and ovalbumin for temperatures below -14°C.

In the following, we concentrate on the IN activity of apoferritin and ferritin samples to find out more about the sites that are responsible for their IN activity.

### 3.2 Freezing experiments performed with apoferritin and ferritin

#### 3.2.1 Batch and concentration dependence

Frozen fractions from two different batches of apoferritin and ferritin with two different concentrations are shown in Fig. 3. For apoferritin (panel a), the higher concentrated solution (0.34 mg/ml) freezes in a narrow temperature range from -4 to -7°C for batch 1 and between -4 and -13°C for batch 2, indicating a higher IN activity of batch 1 than of batch 2. However, for the lower concentration (0.036 mg/ml) the situation is reversed. While the onsets for both batches are at -5°C, batch 2 reaches $FF = 1$ already at -17°C (light blue line) while batch 1 only at (-22°C; dark blue line), indicating a higher IN activity of batch 2 compared to batch 1 for this concentration.

For ferritin (Figure 3b), the difference between batches is even larger. The higher concentrated solution (0.39 mg/ml protein) freezes in the temperature range between -4 and -12°C for batch 1 and between -7.5 and -20°C for batch 2. For the more dilute solutions (0.04 mg/ml), the freezing was observed to occur between -4 and -21°C for batch 1 and between -10 and -22.5°C for

batch 2. This is a remarkable difference in the IN activity between the two batches, putting into question that the fully assembled cage monomer, which is the dominant species present in the solution, is the IN active species.

Therefore, to investigate the influence of the buffer solution and random impurities on IN activity, a portion of apoferritin and ferritin of batches 2 were dialyzed. By means of dialysis, salts and other compounds potentially present in the commercial protein solution are removed. The frozen fraction curves of the dialyzed samples practically overlap for the lower concentrations and show a slight decrease for the higher concentrations for ferritin and a slight increase for apoferritin. The similarity between the IN activity of the dialyzed and the original samples makes it unlikely that random impurities are responsible for the observed IN activity. Overall, the IN activity of ferritin is lower than the one of apoferritin, which makes it unlikely that iron plays an active part in ice nucleation by ferritin. Rather, the presence of iron within the cages seems to reduce the IN activity of the proteins.

To identify the origin of the IN active sites of the ferritin and apoferritin samples, we explored how pH, temperature, and dilution influence their IN activity.

### 3.2.2 pH variations

To test the pH dependence of IN activity, we performed freezing experiments with apoferritin and ferritin samples in buffer solutions with pH from 0 to 9.5. Figure 4 shows the *FF* curves for apoferritin, batch 1, with concentrations of 0.036 mg/ml (panel a) and 0.34 mg/ml (panel c) and for ferritin, batch 1, with concentrations of 0.04 mg/ml (panel b) and 0.39 mg/ml (panel d).

With variations of up to 2°C, the freezing curves of the higher concentrated apoferritin and ferritin samples exhibit only a slight pH dependence in the range from pH 2 to 9.5, with lowest freezing temperatures at pH 7, while pH 2 – 5 curves are shifted to slightly higher temperatures, and the maximum at pH 9.5. The freezing curves of the more dilute samples show the same pH dependence but with a slightly larger spread in temperature. The pH 0 freezing curves are clearly offset to lower temperature but still show high IN activity, which is astonishing considering that protein coagulation was clearly visible in these samples (see Fig. S3 of SI). Apoferritin cages are positively charged below pH 4.0 and negatively charged above pH 4.6 (Petsev and Vekilov, 2000; Valle-Delgado et al., 2005). The conformity in freezing temperatures below and above the isoelectric point at about pH 4 shows that the net charge of apoferritin has no significant influence on IN activity.

Moreover, the ferritin cages undergo conformational changes in the investigated pH range, which also do not seem to influence IN activity strongly. Namely, small-angle X-ray scattering of horse spleen ferritin and apoferritin showed that the apoferritin cage is stable over the pH range from 3.4 to 10 (Kim et al., 2011), but when the pH decreases from 3.40 to 0.80, the cage disassembles stepwise, by first forming a hollow sphere with two holes, then a headset-shaped structure, and finally, rod-like dimers. Disintegration and aggregation of horse spleen ferritin at low and high pH was also observed by Crichton and Bryce (1973), using a sedimentation-velocity technique. They observed that fully assembled cages prevailed for pH values between 2.8 and 10.6, assembled cages and subunits were present at pH = 2.8 – 1.6 and pH = 10.6 – 13.0. At pH = 1.6 – 1.0 subunits were the only identified species, while below pH 1.0 the subunits agglomerated to larger (non-cage) aggregates. The dissociation stops at the level of dimers, since dissociation into subunit monomers does not seem to occur without full denaturation of the protein (Linder et al., 1989). Crichton and Bryce (1973) explained the disassembly at low pH by changes in conformation of apoferritin due to the protonation of carboxyl groups with pKa values of 3.29, initiating the transfer of one tryptophan residue from the interior of the protein to the exterior, hence exposing it to solvent. Subunit dissociation involves the transfer of four to five tyrosine residues to a more hydrophilic environment, most likely to the solvent, accompanied by protonation of at least two carboxyl groups of pKa 2.16. Such changes in subunit conformation likely determine the apoferritin shell disassembly (Santambrogio et al., 1992). Agglomeration of apoferritin and ferritin below pH 1.0 is in accordance with the coagulation in our samples observed at pH 0 (Fig. S3 of SI).

Despite the disintegration of the cages below pH 3.4 and agglomeration below pH 1, the IN activity of apoferritin and ferritin solutions is hardly decreased at pH 2 and still remarkably high at pH 0. This makes it highly unlikely that fully assembled cages are required as the entities that provide IN activity to the apoferritin and ferritin samples. To exclude IN by non-proteinaceous species present as impurities in the ferritin and apoferritin samples, we performed stress tests to determine the stability limit of the ice-nucleating species.

### 3.2.3 Stress treatments

Figure 5 shows the decrease of *FF* depending on the treatment of apoferritin solutions (batch 1, 0.34 mg/ml). While pH 0 decreases the IN activity only by 3°C, heating at 110°C for 5 h shifts freezing by 10 – 15°C to lower temperature. Moreover, it leads to even stronger aggregation of the ferritin and apoferritin samples (see Fig. S4 of SI) than exposure to pH 0. If heating and pH 0 are combined, freezing shifts to the temperature range observed for pure SA water but is still above freezing observed for the pH 0 buffer reference sample. Visual inspection of the vials reveals that both, the ferritin and apoferritin samples are now clear colourless solutions (see Fig. S3 of SI). This shows that low pH and heat needs to be combined to completely disassemble the protein subunits, and to remove the IN activity.

Indeed, ferritin and apoferritin have proved to be very heat resistant. Using UV-Vis spectrophotometry and gel electrophoresis to investigate the thermostability of horse spleen apoferritin, Kudr et al. (2015) report small conformational changes already at 36°C. Above 65°C, the spherical cage structure is lost, accompanied by the release of subunits. This denaturation shows substantial reversibility upon cooling when heating is limited to a few degrees below 68°C. Differential scanning calorimetry attests high thermal stability to horse spleen apoferritin, with denaturation accompanied by aggregation and precipitation occurring only above 93°C under neutral conditions (Stefanini et al., 1996). Evaluation of the enthalpy change suggests that the thermal denaturation does not lead to complete unfolding of the subunits. Moreover, denaturation displays significant reversibility after heating to temperatures only a few degrees below 93°C. Yet, even at 100°C, denaturation of horse spleen ferritin seems incomplete, since the majority of protein spheres appears intact in high resolution electron microscopy, with only a minority being clearly disrupted even after boiling and cooling (Massover,1978). The high thermal stability of ferritin and apoferritin is ascribed to intra- and inter-subunit interactions (Santambrogio et al., 1992; Massover, 1993; Yoshizawa et al., 2007). Thus, IN activity persisting after heating for 5 h at 110°C is in agreement with the high thermal stability of ferritin and apoferritin. This strengthens the assumption that proteinaceous structures are responsible for the IN activity rather than non-proteinaceous impurities. Since horse spleen apoferritin from Sigma-Aldrich should be free of foreign proteins (>99.9 % w/w) (Thomas et al., 1998), the IN activity of the horse spleen apoferritin sample indeed seems to arise from sites connected with apoferritin itself. To elucidate the abundance of such sites and to constrain the size of the IN active entities, we prepared a dilution series for freezing and performed DLS experiments, allowing the correlation of active site densities with the size of apoferritin species present in solution.

### 3.2.4 Apoferritin dilution series

We diluted the apoferritin sample (batch 2) in steps of factors of 2 to 3, until the freezing curve was close to the one of pure SA water. Figure 6 shows the freezing curves covering concentrations from 0.34 mg/ml to 0.56 μg/ml (0.7 – 0.001 μM). The sample with the highest concentration is identical with batch 2 apoferritin shown in Fig. 2 with onset at -4°C and *FF* = 1 at -13°C. Dilution to 0.0045 mg/ml decreases *FF* above -11°C from 1 to 0.1, while even further dilution reduces freezing below -11°C, indicating the presence of two distinct freezing ranges. This division of freezing into two distinct ranges is also visible in the differential active site densities calculated using Eq. 2 (panel b), which display almost constant values of 10 to 100 mg$^{-1}$K$^{-1}$ between -5 and -11°C, followed by a steep increase to almost 1000 mg$^{-1}$K$^{-1}$ between -11 and -15°C, and a shallower increase when temperature is further decreased to -22°C. The division into two distinct freezing ranges is also visible in the cumulative active site densities calculated with Eq. 1 (panel c). Moreover, it can be seen that the active site densities between

-4°C and -11°C for apoferritin concentrations of 0.036 mg/ml and 0.009 mg/ml are slightly higher than for the other concentrations and that the active site densities for the lowest concentration of 0.56 μg/ml feature a strong decrease between -17 and -21°C. However, the large contribution of SA water to the frozen fraction for this apoferritin concentration make the active site density originating from apoferritin subject to large uncertainties.

To relate the freezing temperature to the size of apoferritin species present in the sample, we performed DLS measurements. Figure 7 presents the hydrodynamic diameter of apoferritin (batch 2) species present in the solution for concentrations from 0.34 mg/ml to 0.0045 mg/ml. At lower concentrations, no reproducible DLS curves could be obtained due to the high dilution. The size distribution is strongly dominated by the major peak at 12.5 ± 0.3 nm. Moreover, there are two additional weak peaks with maxima around 500 nm and 5000 nm that are shown on an enlarged scale in Fig. 7, suggesting the presence of larger

aggregate species. The peak maximum at 12.5 ± 0.3 nm (taken as the average of the peak maxima from 0.34 mg/ml to 0.009 mg/ml) agrees with the reported hydrodynamic diameter of the apoferritin cage of 12.7 nm (Petsev et al., 2000; 2001). This value confirms the presence of cage monomers as the dominant species. Nevertheless, the presence of cage dimers with a hydrodynamic diameter of 18.4 nm (Petsev et al., 2001) and trimers is also likely since these species would not be resolved from the monomers by DLS. According to Richter and Walker (1967), cage dimers, trimers and oligomers are in dynamic

equilibrium with cage monomers and become abundant at high solution concentrations (>2 mg/ml). In addition, there is a fraction of oligomeric species that are stabilized by partial unfolding of some of the apoferritin subunits which leads to the exposure of hydrophobic parts of the protein to the water environment resulting in attraction between cage monomers instead of repulsion (Yang et al., 1994; Petsev, 2000; Petsev et al., 2000; 2001). These oligomers do not dissociate into cage monomers when the sample is diluted. Thomas et al. (1998) analysed horse spleen apoferritin as received from Sigma-Aldrich and found

in addition to monomers (M ≈ 440 kDa), dimers (M ≈ 880 kDa), and trimers (M ≈ 1300 kDa) also other species, namely, intermediate oligomers (880 kDa < M < 1300 kDa), larger oligomers (M > 1300 kDa), intermediate aggregates (~10 mer with M ≈ 5000 kDa) and large aggregates (180 mer with M ≈ 80 MDa), free subunits with M < 67 kDa and also some low molecular weight species with M ≈ 14 kDa and 6 kDa likely due to proteolysis.

As shown in Fig. 8a, the monomer peak maximum is constant within error for concentrations down to 0.009 mg/ml, but it

significantly drops to 8.9 ± 1.3 nm for the lowest concentration that could be investigated with DLS (0.0045 mg/ml). The decrease in size is likely due to the dissociation of cage monomers into subunits. The maximum at 8.9 nm points to subunit hexamers (quarter cages) or subunit dodecamers (half cages), yet, the presence of subunit dimers and whole cage monomers is also likely. Indeed, with high resolution electron microscopy, Massover (1980) detected small objects below the size of cage monomers when the ferritin concentration was less than 0.01 μg/ml in agreement with a dynamic equilibrium between ferritin

dissociation and association. It can be assumed that upon further dilution, this dissociation progresses until subunits, mostly dimers prevail, (Massover, 1980; Massover, 1993).

Figure 8b displays the volume fraction of aggregates as a function of solution concentration by summing over the diameter range from 68 – 10$^4$ nm of the DLS volume-weighted distribution. It shows that the ratio between monomeric/oligomeric species and aggregates is not constant but shifts slightly to aggregates for intermediate concentrations from 0.036 to 0.009

mg/ml. This shift brings about a slight increase in aggregate concentration when the apoferritin concentration is decreased from 0.068 mg/ml to 0.036 mg/ml (see Fig. 8c). A reason for the increased aggregation might be that we diluted the original apoferritin solution in NaCl with pure water. Since aggregation depends on repulsion between apoferritin cages which is influenced by the presence of electrolytes (Petsev, 2000; Petsev et al., 2000; 2001; Manciu and Ruckenstein, 2002), dilution could have promoted aggregation. The solutions with increased concentrations of aggregates (0.036 to 0.009 mg/ml) show at

the same time the largest variability of aggregate concentration between measurements (note error bar lengths in Figs. 8b and c).

The shift to cage aggregates (cage multimers) correlates with the slightly increased active site densities in the temperature range from -4 to -11°C observed for these concentrations in the freezing experiments (see Figs. 6b and 6c). We take this as

evidence that ice nucleation in the temperature range between -4 and -11°C stems from active sites on aggregates. Since the loss of IN activity between -11°C and -21°C starts at 0.009 mg/ml and goes along with the decrease of monomeric/oligomeric species, we consider sites on cage monomers or oligomers responsible for ice nucleation in this temperature range. This evidences the importance of the protein assembly for IN activity and the relevance of aggregation to reach high freezing temperatures. In the following, the role of protein assembly will be further elucidated by disassembly-reassembly experiments.

### 3.2.5 Apoferritin disassembly-reassembly

Apoferritin cages have been shown to disassemble below pH 3 into subunit dimers and reassemble within 10 to 40 min when pH is raised again to 4 – 8, as illustrated in Fig. 9c (Gerl and Jaenicke, 1987; 1988; Smith-Johannsen and Drysdale, 1969, Linder et al., 1989). To further investigate which apoferritin species are relevant for IN activity, disassembly-reassembly experiments were carried out for two different apoferritin concentrations. Figure 9a shows the frozen fraction as a function of temperature for apoferritin, batch 2, 0.036 mg/ml. Due to cage disassembly at pH 2, the freezing temperature decreases by about 2°C in the temperature range from -11 to -21°C, but only by about 0.5°C close to the freezing onset at about -5°C. Cage reassembly at pH 8 fully restores the IN activity between -11 and -21°C but induces hardly any increase between -4 and -11°C. Similarly, the experiments performed at lower concentration (0.018 mg/ml, panel b) show a reversible decrease in IN activity between -11 and -21°C and no decrease between -4 and -11°C at pH 2.

To identify the species present after disassembly at pH 2, DLS measurements of the pH 2-buffered apoferritin solutions were performed. Figure 10 presents the volume-weighted distribution of apoferritin (batch 2) with 0.036 mg/ml (panel a) and 0.018 mg/ml (panel b) for the directly prepared solutions and the disassembled ones at pH 2. For 0.036 mg/ml the cage disassembly appears in the DLS measurements as a shift of the main peak from $12.6 \pm 0.8$ nm to $5.8 \pm 1.3$ nm. For 0.018 mg/ml the main peak shifts from $12.0 \pm 1.0$ nm at neutral conditions to $6.8 \pm 1.7$ nm at pH 2. A diameter of about 6.5 nm corresponds to subunit dimers (Kim et al., 2011). The broad peak width indicates the presence of a wide distribution of subunit species up to cage monomers, and possibly even cage dimers and cage trimers. The aggregate peaks show large variations between replicate measurements.

Considering the large standard deviations, there is no clear difference in aggregate volume fraction between directly prepared and disassembled samples for both investigated concentrations. These results confirm that cage monomers or small cage oligomers are responsible for the heterogeneous freezing observed between -11 to -21°C and cage aggregates for freezing between -4 and -11°C.

### 3.2.6 Ferritin species present in solution

To further substantiate that aggregates are responsible for the IN activity of ferritin and apoferritin above -11°C, we performed DLS measurements with ferritin, batch 2, concentrations of 0.04 mg/ml and 0.39 mg/ml. For these solution concentrations, the freezing experiments featured freezing onset temperatures of -10 and -8°C, respectively (see Fig. 3). The volume-weighted size distributions shown in Fig. 11 disclose an aggregate peak above 1000 nm but lack the aggregate peak below 1000 nm, which is present in the DLS measurements performed with apoferritin. This suggests that aggregates with sizes below 1000 nm are relevant for the IN activity of ferritin and apoferritin above -11°C. These findings support our interpretation that apoferritin cage aggregates are responsible for freezing between -4°C and -11°C, while cage monomers and small cage oligomers are relevant for IN activity below -11°C. To test how stable aggregates are and whether active sites persist during freeze-thaw cycles we performed in the following refreeze experiments.

### 3.2.7 Refreeze experiments

We repeated freeze–thaw cycles in refreeze experiments with the same filling of the 96-well tray for three different apoferritin concentrations (batch 2, 0.34 mg/ml, 0.036 mg/ml and 0.018 mg/ml). For each concentration we prepared three independent fillings, and for each filling, we performed a first freeze-thaw cycle followed by four refreeze cycles. Figure 12 shows the frozen fraction as a function of temperature in the top three rows and the evolution of the frozen fraction for selected temperatures (-5, -10, -15, and -20°C) with increasing number of freeze-thaw cycles (bottom row). While the frozen fraction at -5°C slightly increased with increasing number of freeze-thaw cycles, it rather decreased for T = -15°C and -20°C. This indicates that additional aggregate sites emerge with increasing number of freeze-thaw cycles (or increasing time in water), while monomeric/oligomeric sites tend to disappear. For the highest investigated concentration (0.34 mg/ml) almost all wells froze in the temperature range from -4 to -11°C due to active sites on aggregates. For the two lower concentrations (0.036 mg/ml and 0.018 mg/ml) some wells froze between -4 and -11°C while the majority froze in the temperature range indicative for monomeric or oligomeric sites (-11 to -21°C). Note, that the freezing events at the lowest temperatures (below -21°C) might also arise from random impurities present in SA water.

To assess the stability of active sites, we analysed the refreeze experiments on a well per well basis. Figure 13 displays the freezing temperatures per well for the 2$^{nd}$ refreeze experiment with apoferritin concentration of 0.036 mg/ml (similar plots for the other experiments are shown in Fig. S5 of the SI). A detailed analysis of all refreeze experiments carried out with 0.036 mg/ml and 0.018 mg/ml apoferritin is presented in Tables 2 and 3, respectively. For the three refreeze experiments performed with 0.036 mg/ml apoferritin samples, *FF* at -11°C (averaged over all 96 wells and the five cycles) was 0.219 for the 1$^{st}$ experiment (i.e. 29 wells were frozen at -11°C in the first cycle, 19 in the second, 20 in the third, 17 in the fourth, and 20 in the fifth cycle, yielding $(29 + 19 + 20 + 17 + 20)/(5 \times 96) = 0.219$), 0.304 for the 2$^{nd}$ experiment, and 0.319 for the 3$^{rd}$ experiment . Assuming that freezing temperatures are stochastic without any dependence on the specific well, the probability that a well constantly froze at T > -11°C is therefore 0.002 – 0.01 ($0.219^4$ to $0.319^4$). Thus, the fraction of wells that always froze at T > -11°C should be 0.002 – 0.01. However, evaluation of the well by well results (shown in Figs. 13 and S5) yielded fractions of wells always freezing at T > -11°C from 0.156 (i.e. in the first experiment, 15 out of the 96 wells froze above -11°C during all five cycles) to 0.271 (see Table 2), indicating that freezing sites on aggregates tend to persist over several freeze-thaw cycles. Similar conclusions can be drawn from refreeze experiments performed with apoferritin concentrations of 0.018 mg/ml with *FF* at -11°C of 0.129, 0.150, and 1.65. Assuming again random occurrence of freezing, the probability that a well always froze above -11°C would equal 0.0003 – 0.0007 ($0.129^4$ to $1.65^4$), implying that no well should freeze constantly above -11°C. Nevertheless, we found a fraction of 0.083 to 0.146 (see Table 3). On the other hand, freezing events of some wells span temperature ranges of over 15°C, indicating that some aggregate active sites appeared or disappeared over a sequence of five freeze-thaw cycles. Overall, the spread of freezing temperature between freeze-thaw cycles of one well is distinctly smaller than the spread over the whole plate, indicating that ice nucleation does not occur on frequent apoferritin sites each with a low IN probability, but on few sites that induce freezing with a high probability and persist over several freeze-thaw cycles.

### 3.3 Comparison with other ice-nucleating proteins

Sect. 3.1 revealed IN activity of all screened proteins and the virus. This outcome is astonishing, considering that so far IN activity has only been found in some bacteria, yeasts, lichen, and fungi, which express IN active proteins tailored for the purpose to nucleate ice. Instead, the investigated proteins have very diverse functions, which are not ice nucleation.

The IN activity of the ice-nucleating protein expressed by *P. syringae* deteriorates outside the pH range from 6 to 8 and decreases by about 6°C after a 10-min heat treatment at 40°C (Pouleur et al., 1992). In contrast, the IN activity of apoferritin and ferritin showed little variation from pH 1 – 9 and was also heat-resistant. This insensitivity to heat and pH resembles that

of the IN active fungal species *Fusarium avenaceum*, which persists up to -2.5°C and is also of proteinaceous nature. IN activity of *F. avenaceum* remains constant from pH 1 to 13 and proves heat tolerant up to 60°C (Pouleur et al., 1992). Moreover, it is preserved after passing through a 0.22 µm-pore-size filter, indicating that the IN active proteins are not bound to a cell membrane (Pouleur et al., 1992).

Using radiation inactivation analysis, Govindarajan and Lindow (1988) found a minimum mass of 150 kDa for IN active sites of *P. syringae* with activity at -12° to -13°C, in agreement with the apparent mass of the IN active proteins expressed by *P. syringae* (Lindow et al., 1989). Apoferritin cage monomers (~480 kDa), cage dimers (~960 kDa) and cage trimers (~1440 kDa) with activity between -11°C and -21°C, exhibit by a factor 3 – 9 times larger masses, indicating that the apoferritin structure is less optimized for ice nucleation than the one of the ice-nucleating protein of *P. syringae*. For IN activity at -2°C,

Govindarajan and Lindow (1988) determined a mass of 19'000 kDa, arising through aggregation of the proteins on the outer membrane of intact cells (Yankofsky et al., 1981). Qiu et al. (2019) also found that aggregation increased the ability of ice-binding proteins to induce ice nucleation. This is consistent with our finding that aggregates are responsible for the IN activity at higher temperatures. Despite their low molecular masses ranging from 10 to 67 kDa, all the proteins screened in this study were able to induce freezing up to at least -8°C. This hints to oligomers or aggregates of these proteins as the IN active species.

Indeed, casein, which is known to form micelles (Dalgleish and Corredig, 2012) consistently induced freezing at high temperatures (from -8 to -13°C). In contrast, the hydrophobins (HPA and HPB), which form monolayer coatings on surfaces rather than aggregates exhibit only a small fraction of sites that are active between -6 and -8°C. The ice-binding protein LeIBP with only few nucleation events above -10°C is known to dimerize in solution (Lee et al., 2012), but might have a low tendency to form larger aggregates.

The apoferritin dilution series covers the range from highly IN active at 0.34 mg/ml to similarly IN active as pure water at 0.56 µg/ml. Given a sample volume of 50 µl, the number of apoferritin cages ranges from $2.1 \cdot 10^{13}$ to $3.5 \cdot 10^{10}$ per well, indicating that IN activity is limited to a tiny fraction of the monomeric/oligomeric species, and probably to only a minority of aggregates. In the case of *P. syringae* the fraction of IN active cells strongly varies, but is clearly higher than the number of IN active sites in apoferritin. Depending on cultivation, every tenth to one cell in a million show IN activity at -2 to -4°C. (Déspres et al.,

2012; Murray et al., 2012). Snomax®, a commercial product containing IN active proteins from non-viable *P. syringae* exhibits active site densities increasing from $10^4$ to $10^9$ mg$^{-1}$ for temperatures decreasing from -5°C to -10°C (Wex et al., 2015; Kanji et al., 2017). In comparison, the apoferritin active site density per mass is small, ranging from 1 mg$^{-1}$ at -5°C to 100 mg$^{-1}$ at -10°C, again pointing to a low density of IN active sites occurring accidentally on apoferritin. Interestingly, the IN activity of Snomax® decreases with storage time indicating that the most efficient nucleation sites of Snomax® degrade with time (Polen

et al., 2016; Häusler et al., 2018). This may be due to loss of free hydrogen bonding sites or disintegration of larger aggregates. Ice-nucleating proteins expressed by Pseudomonas contain a central domain, composed of 50–80 repeats of 16 amino acids with the sequence GYGSTxTAxxxSxLxA where x can be any amino acid. The repetition of this sequence is considered to provide the ice-templating sites (Ling et al., 2018; Wolber, 1993; Schmid et al., 1997) by binding water molecules to the threonine-x-threonine (TxT) motif of this sequence, and thus aligning water molecules into a favorable pattern for the

formation of an ice embryo (Garnham et al., 2011; Graether and Jia, 2001). Apoferritin lacks repetition units containing the TxT motif (Andrews et al., 1992), showing that IN activity of proteins can also arise from other structures.

Recently, antifreeze and ice-binding proteins have proved to be IN active when they aggregate to larger structures, which was explained by ice-nucleating sites that emerge when ice-binding sites are repeated in aggregated structures (Eickhoff et al., 2019; Hudait et al., 2018; Qiu et al., 2019). Nevertheless, the IN activity of the ice-binding protein LeIBP is only intermediate,

compared with the other proteins screened in Sect. 3.1, although LeIBP exhibits a repetition sequence that allows it to bind to ice. Our finding of a general ability of proteins to nucleate ice indicates that a repetition section matching the ice structure is not a prerequisite for IN activity in proteins.

In a recent study, Pandey et al. (2016) demonstrated by sum frequency generation (SFG) spectroscopy and molecular dynamics simulations that the ice-active sites of *P. syringae* feature hydrophilic-hydrophobic patterns that enhance ice nucleation. Moreover, time-resolved SFG spectroscopy showed that the protein facilitates the removal of latent heat from the nucleation site. Since the screened proteins all have characteristic freezing onset temperatures, their nucleation sites do not seem to be totally random but related to the protein structure. A templating effect may result from the pattern of hydrophilic and hydrophobic regions on alpha helices and beta sheets together with sites for hydrogen bonding responsible for the tertiary and quaternary structure. In misfolded proteins, these may be available to bind water molecules. Attached to ferritin are water molecules in inter-subunit interfaces through hydrogen bonds (Hempstead et al., 1997), which may be a starting point for ice embryos. Also, the outer protein shell features iron bonding sites (Massover, 1993), which may play a role in ice nucleation. The refreeze experiments have shown that IN activity of apoferritin is localized in few sites of high IN efficiency while the rest of the proteins is inactive. Our screening showed that such highly active sites seem to arise in proteins with very different functions. This indicates that protein structures have an inherent ability to nucleate ice that can be evoked through misfolding of the amino acid chains or through aggregation.

## 3.4 Atmospheric implications

Different common proteins showed IN activity in DRINCZ experiments with freezing onsets above -10°C. This supports the potential relevance of biological INP for ice formation at mixed-phase cloud conditions. Ferritin and apoferritin, the largest among the investigated proteins showed IN activity up to -4°C, yet, with a low density of active sites.

Since ferritins are present in most organisms including bacteria, animals and plants, they might well be released to the environment after the death of organisms when cells are disrupted. However, our measurements indicate that the IN activity disappears in highly diluted solutions, most probably because aggregated ferritin and apoferritin disintegrate and cages disassemble into subunits. Indeed, if aggregation were a prerequisite for ice nucleation in proteins, the effect of dilution might reduce the ice-nucleation potential of proteins in general. Moreover, different types of proteins will be mixed with each other and with organic and non-proteinaceous biological material leading to mixed aggregates in aerosol particles. Therefore, in order to be of atmospheric relevance, mixed aggregates also need to have the ability to nucleate ice. While proteins may aggregate in aerosol particles, they need to be prevented from disintegrating when aerosol particles dilute during cloud droplet activation. Therefore, to stick together, they might need to adhere to a surface, e.g. the droplet surface or mineral surfaces. Mineral surfaces keeping proteins aggregated might explain findings that soil dust containing minerals together with biological material is able to nucleate ice at higher temperatures than dust aerosols from deserts (Pratt et al., 2009; Conen et al., 2011; O'Sullivan et al., 2014; Tobo et al., 2013; 2014; Augustin-Bauditz et al., 2016).

## 4 Conclusions

Freezing experiments performed with horse spleen ferritin and apoferritin reveals IN activity in two distinct temperature ranges, namely from -4 to -11°C and from -11 to -21°C. We exposed the samples to different conditions to identify the nature of the IN active entities:

− The strong reduction of freezing temperatures after combined acid and heat treatment indicates that proteinaceous species are responsible for the observed IN activity.

− The resistance of IN activity to heat treatment (5 h at 110 °C) corresponds to the high thermal stability of ferritin and apoferritin.

− At concentration below 0.56 µg/ml the frozen fraction of the horse spleen apoferritin sample reached similarly low values as the SA water background. At this concentration more than $10^{10}$ cage monomers are still present in each 50 µl sample aliquot.

Taking these findings together, the IN activity seems to stem from proteinaceous species but not from the regularly folded cage monomers. Indeed, horse spleen apoferritin solutions contain apart from the dominating cage monomers also cage aggregates, misfolded cage monomers, as well as oligomeric species such as cage dimers and cage trimers. Correlating DLS measurements with freezing results indicates that ferritin and apoferritin aggregates are responsible for the IN activity between

-4 and -11°C, and misfolded monomeric or oligomeric species between -11 and -21°C

Batch to batch variability of aggregate and oligomer concentrations may also explain the observed variation in *FF* between the investigated batches of ferritin and apoferritin. Moreover, the lower IN activity of ferritin compared to apoferritin suggests that the iron oxide plays no active role in the IN activity of ferritin.

The apoferritin results together with the screening experiments performed with different proteins indicate that IN activity is a

common property of proteins most probably arising accidentally through favorable aggregation of single proteins (or single protein cages) into larger structures. However, it is questionable whether this accidental, proteinaceous IN activity is of relevance in the atmosphere for mixed-phase cloud glaciation, since protein aggregates can disintegrate due to dilution in cloud droplets. Yet, if proteins aggregated at droplet or mineral surfaces were IN active, this might provide an explanation for the IN activity of biological material and the superior IN activity of soil dust compared with mineral dust.

*Data availability.* The data presented in the figures is available at the ETH repository: https://doi.org/10.3929/ethz-b-000391914.

*Author contributions.* MCC conducted the ice nucleation experiments. ROD introduced MCC to the instrument and supported

her with the measurements. MCC, ROD, MAIA, AMB, and CM contributed to the planning and interpretation of the experiments. MAIA conducted the DLS measurements and prepared the figures. CM prepared the manuscript with contributions from MAIA, MCC, ROD, and AMB.

*Competing interests.* The authors declare that they have no conflict of interest.

*Acknowledgements.* We acknowledge the Swiss National Foundation for financial support (project numbers: IZSEZ0_179149/1 and 200021_156581). We are grateful for financial support from the Basque Government (Elkartek programmes ng 15 and ng 17), and from the Spanish MINECO (MAT2013-46006-R grant, Red Temática de Excelencia en Física Virológica, Maria de Maeztu "Units of Excellence" Programme MDM-2016-0618). We thank Ulrike Lohmann and

Zamin Kanji for helping to initiate and supporting this project, Nadine Borduas-Dedekind and Julia Werz for sample preparation optimization and preliminary results, María Isabel Asenjo Sanz (Centro de Física de Materiales) and Jorge Melillo (Centro de Física de Materiales, CIC nanoGUNE) for valuable discussions and assistance during the DLS measurements. We acknowledge Mitsuhiro Okuda, Zamin A. Kanji and Nadine Borduas-Dedekind for helpful discussions. We thank Thomas Subkowski (White Biotechnology Research, BASF) for supplying the hydrophobins, and Christina Wege from the University

of Stuttgart for contributing the tobacco mosaic virus. We are especially thankful to Dr. Se Jong Han from the Korea Polar Research Institute (KOPRI) for providing the ice-binding protein LeIBP.

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

**Table 1** pH values of the apoferritin and ferritin solutions in water and the buffered samples.

| Solution/buffer composition | pH | | | |
|---|---|---|---|---|
| | Apoferritin | | Ferritin | |
| | 0.34 mg/ml | 0.036 mg/ml | 0.39 mg/ml | 0.04 mg/ml |
| Sigma-Aldrich water | 7.00 | 6.97 | 7.60 | 7.27 |
| 85.8 mM HCl | 0.27 | 0.16 | 0.02 | 0.07 |
| 50 mM KCl + 10.6 mM HCl | 2.08 | 2.07 | 2.08 | 2.06 |
| 71.5 mM citric acid + 57 mM $Na_2HPO_4 \cdot 7H_2O$ | 3.52 | 3.47 | 3.67 | 3.46 |
| 48.5 mM citric acid + 103 mM $Na_2HPO_4 \cdot 7H_2O$ | 5.27 | 5.07 | 5.14 | 5.06 |

| | | | | |
|---|---|---|---|---|
| 0.5 mM HEPES (adjusted with 5M NaOH) | 7.06 | 7.11 | 7.07 | 7.12 |
| 12.5 mM $Na_2B_4O_7 \cdot 10H_2O$ + 3.6mM NaOH | 9.50 | 9.51 | 9.49 | 9.48 |

**Table 2** Freezing statistics of refreeze experiments performed with apoferritin solutions, batch 2, 0.036 mg/ml

|  | 1st experiment | 2nd experiment | 3rd experiment |
|---|---|---|---|
| Freezing temperature range over all wells and all cycles | -5.1 – -20.4°C | -5.1 – -19.5°C | -4.3 – -19.7°C |
| Smallest freezing range of a well over all cycles | -8.8 – -9.7°C | -12.7 – -13.4°C | -11.8 – -12.2°C |
| Largest freezing range of a well over all cycles | -6.5 – -17.0°C | -5.6 – -18.8°C | -5.1 – -16.1°C |
| Average freezing temperature of all wells over all cycles | -12.5°C | -11.9°C | -11.5°C |
| Highest freezing temperature of a well over all cycles | -6.6°C | -5.5°C | -4.9°C |
| Coldest freezing temperature of a well over all cycles | -17.5°C | -17.4°C | -16.9°C |
| Frozen fraction at $T = -11$°C | 0.219 | 0.304 | 0.329 |
| Fraction of wells always frozen at $T = -11$°C | 0.156 | 0.219 | 0.271 |

**Table 3** Freezing statistics of refreeze experiments performed with apoferritin solutions, batch 2, 0.018 mg/ml

|  | 1st experiment | 2nd experiment | 3rd experiment |
|---|---|---|---|
| Freezing temperatures range over all wells and all cycles | -4.8 – -24.5°C | -4.1 – -24.0°C | -5.0 – -23.5°C |
| Smallest freezing range of a well over all cycles | -16.6 – -17.1°C | -12.0 – -12.7°C | -18.3 – -19.3°C |
| Largest freezing range of a well over all cycles | -8.6 – -22.8°C | -5.3 – -21.4°C | -6.5 – -17.3°C |
| Average freezing temperature of all wells over all cycles | -14.0°C | -14.2°C | -14.2°C |
| Highest freezing temperature of a well over all cycles | -5.5°C | -4.1°C | -6.5°C |
| Coldest freezing temperature of a well over all cycles | -19.9°C | -19.9°C | -22.1°C |
| Frozen fraction at $T = -11$°C | 0.165 | 0.150 | 0.129 |
| Fraction of wells always frozen at $T = -11$°C | 0.146 | 0.094 | 0.083 |

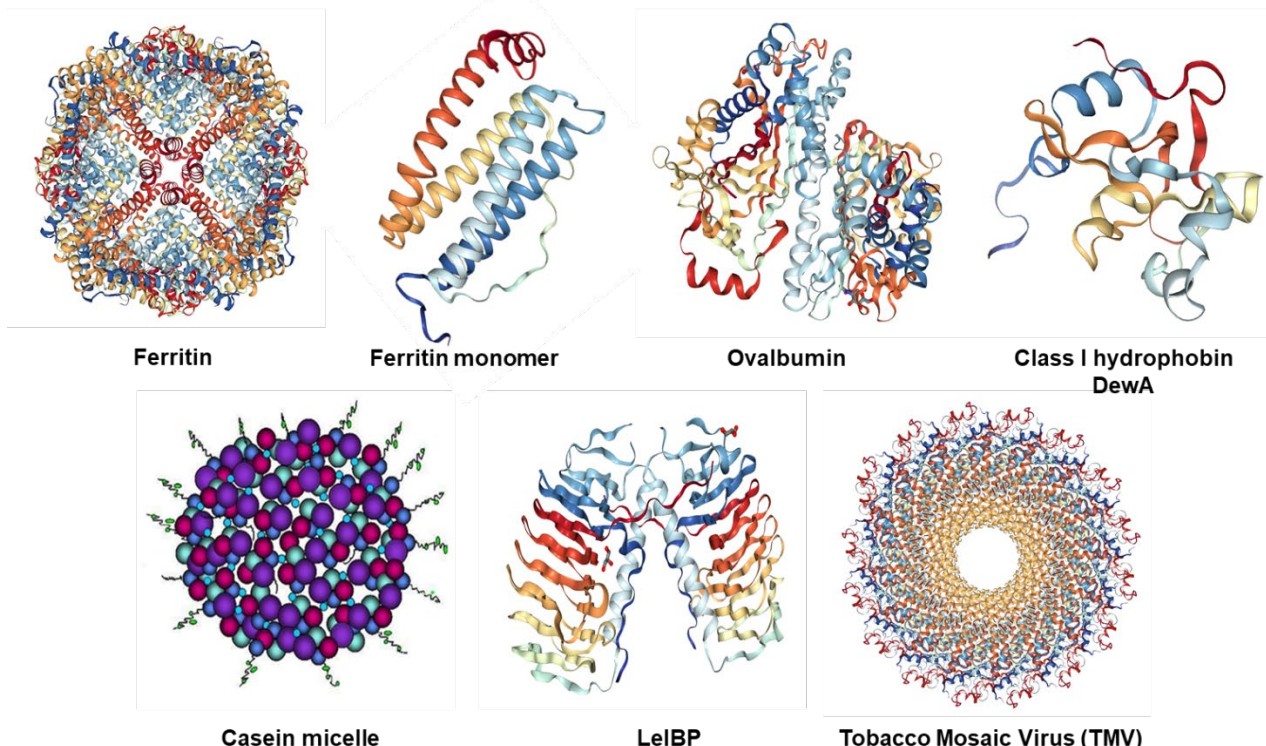

**Figure 1.** Structures of the investigated proteins and the tobacco mosaic virus: Horse spleen (apo)ferritin cage and (apo)ferritin monomer (PDB ID: 4V1W, Russo and Passmore, 2014), chicken ovalbumin (PDB ID: 1OVA, Stein et al., 1991), hydrophobin class I DewA (PDB ID: 2LSH; Morris et al., 2013b), casein micelle consisting of the four different casein proteins: $\alpha_{s1}$-casein (purple), $\alpha_{s2}$-casein (light blue), $\beta$-casein (red), and $\kappa$-casein (dark blue with tail) (adapted from Rebouillat and Ortega-Requena, 2015), ice-binding protein LeIBP dimer (PDB ID: 3UYU, Lee et al. 2012), and tobacco mosaic virus (TMV) (PDB ID: 3J06; Ge and Zhou, 2011).

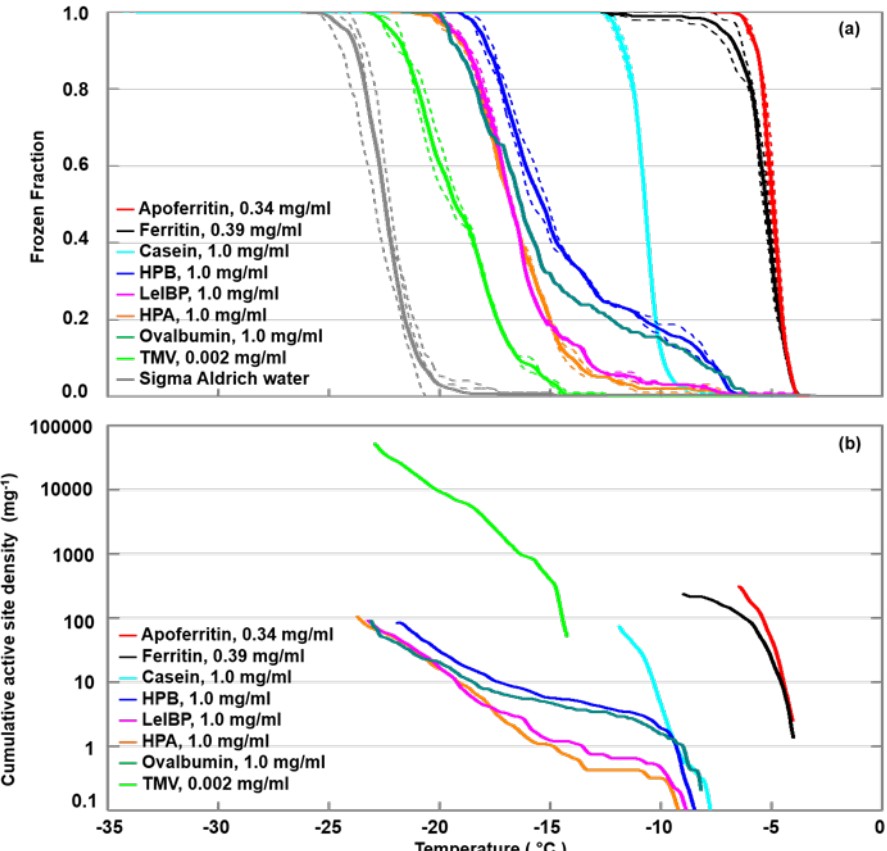

**Figure 2.** DRINCZ experiments performed with apoferritin (0.34 mg/ml, batch 1), ferritin (0.39 mg/ml, batch 1), casein (1.0 mg/ml), HPA (1.0 mg/ml), HPB (1.0 mg/ml), LeIBP (1.0 mg/ml), ovalbumin (1.0 mg/ml), and TMV (0.002 mg/ml). Panel a: frozen fraction as a function of temperature. For comparison, freezing of pure SA water is also shown. Single freezing runs are shown as dashed lines and the mean as the thick solid lines. Panel b: Cumulative active site densities.

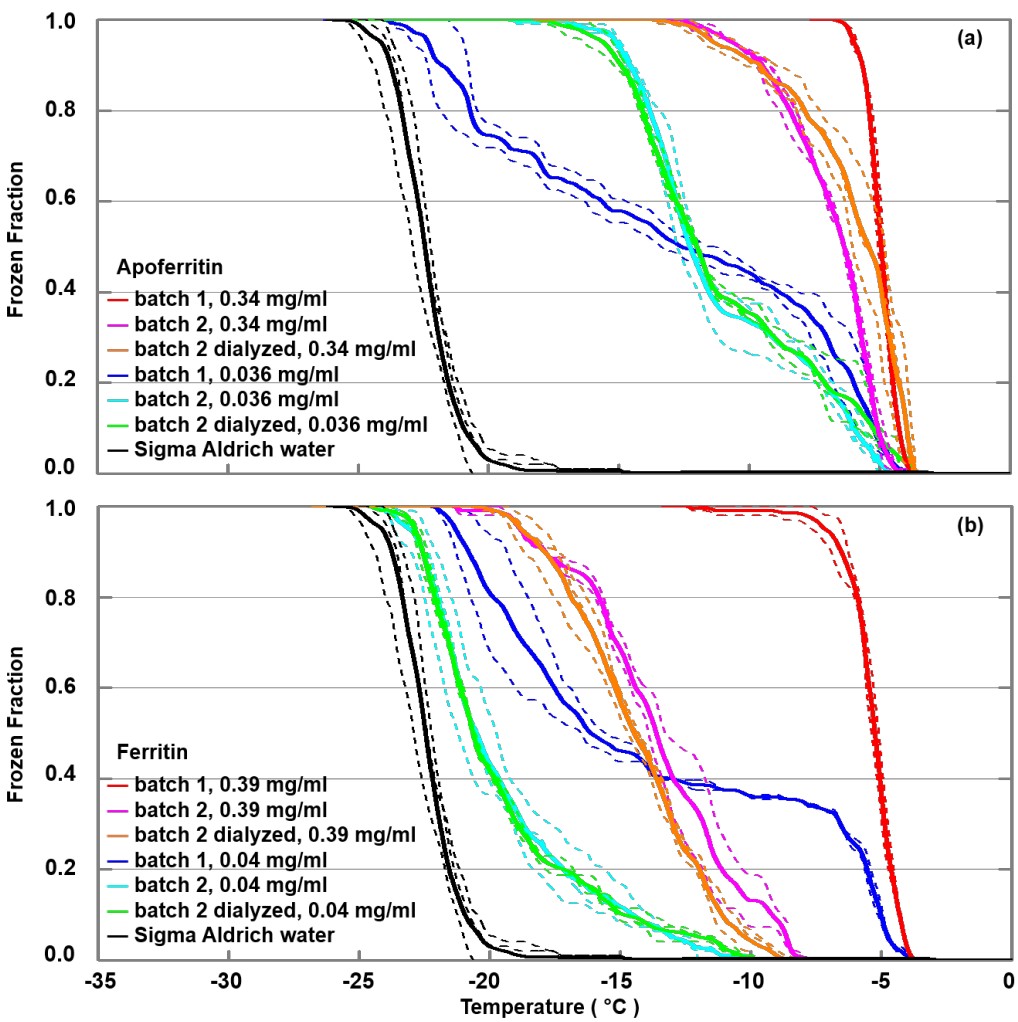

**Figure 3.** Batch dependence of the IN activity of apoferritin and ferritin for two different concentrations. (a) Frozen fraction as a function of temperature for batches 1 and 2 of apoferritin solutions with concentrations of 0.34 mg/ml and 0.036 mg/ml) and a dialyzed solution of batch 2 prepared with the lower concentration (see material and methods for details). (b) Frozen fraction as a function of temperature for batches 1 and 2 of ferritin solutions with concentrations of 0.39 mg/ml and 0.04 mg/ml) and a dialyzed solution from batch 2 prepared with the lower concentration. Two to three DRINCZ experiments were performed for each concentration (dashed lines) and the mean is shown as the thick solid lines.

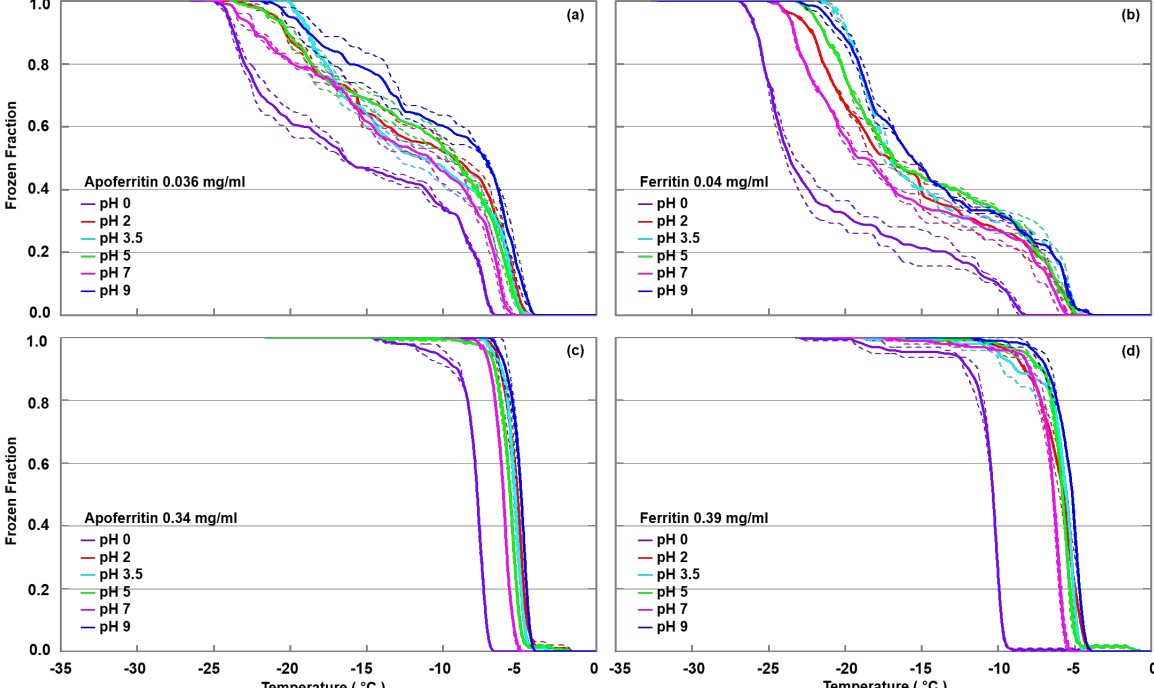

**Figure 4.** pH dependence of frozen fractions at six different pH for apoferritin (batch 1, 0.036 mg/ml) (panel a), ferritin (batch 1, 0.04 mg/ml) (panel b), apoferritin (batch 1, 0.34 mg/ml) (panel c), ferritin (batch 1, 0.39 mg/ml) (panel d). Two to three DRINCZ experiments were performed for each concentration (dashed lines) and the means are shown as the thick solid lines.

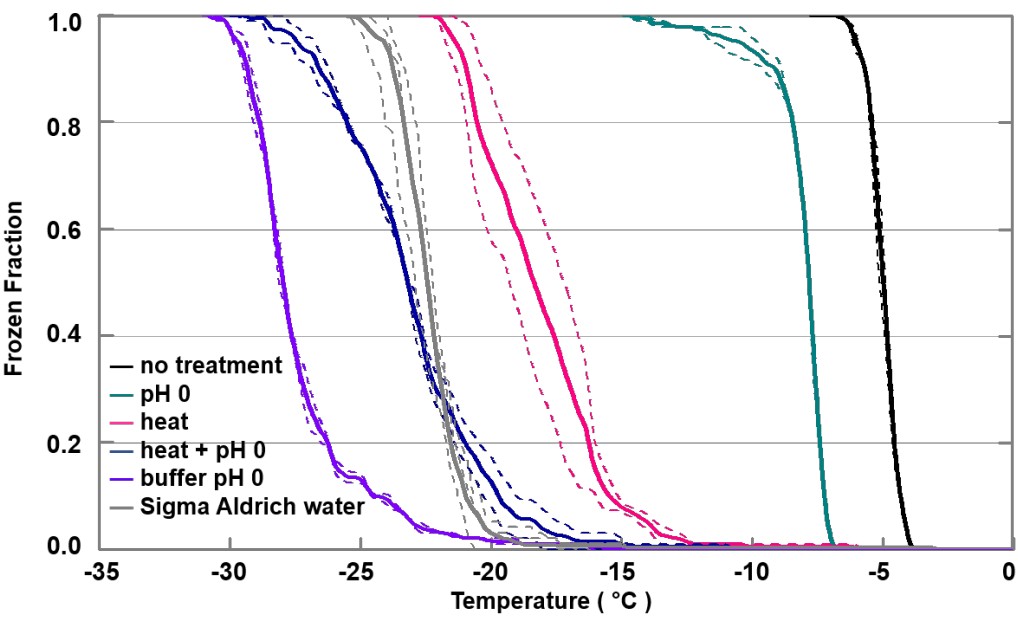

**Figure 5.** Stress treatments performed with apoferritin batch 1 (0.34 mg/ml). Frozen fraction as a function of temperature for apoferritin in pH 0 buffer, after heating an apoferritin solution in SA water for 5 h at 110°C, and after heating an apoferritin solution at pH 0 for 5 h at 110°C. For comparison, freezing of SA water and of the buffer solution at pH 0 is also shown. Two DRINCZ experiments were performed for each concentration (dashed lines) and the mean is shown as the thick solid lines.

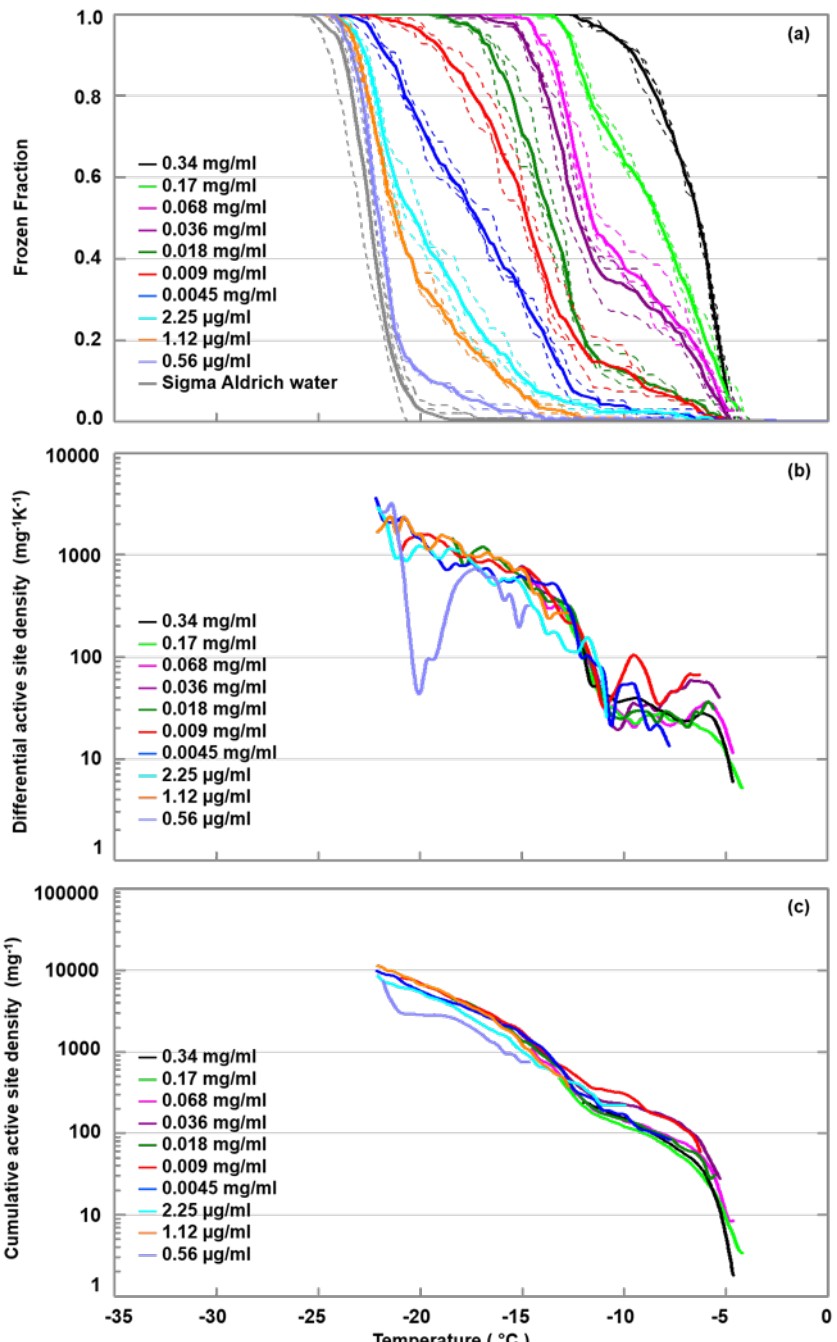

**Figure 6.** Freezing experiments with an apoferritin (batch 2) dilution series. (a) Fraction of droplets frozen as a function of temperature for concentrations between 0.34 mg/ml and 0.56 µg/ml. (b) Differential IN active site densities per unit mass and Kelvin calculated with Eq. 2. (c) Cumulative active site densities per unit mass calculated with Eq. 1.

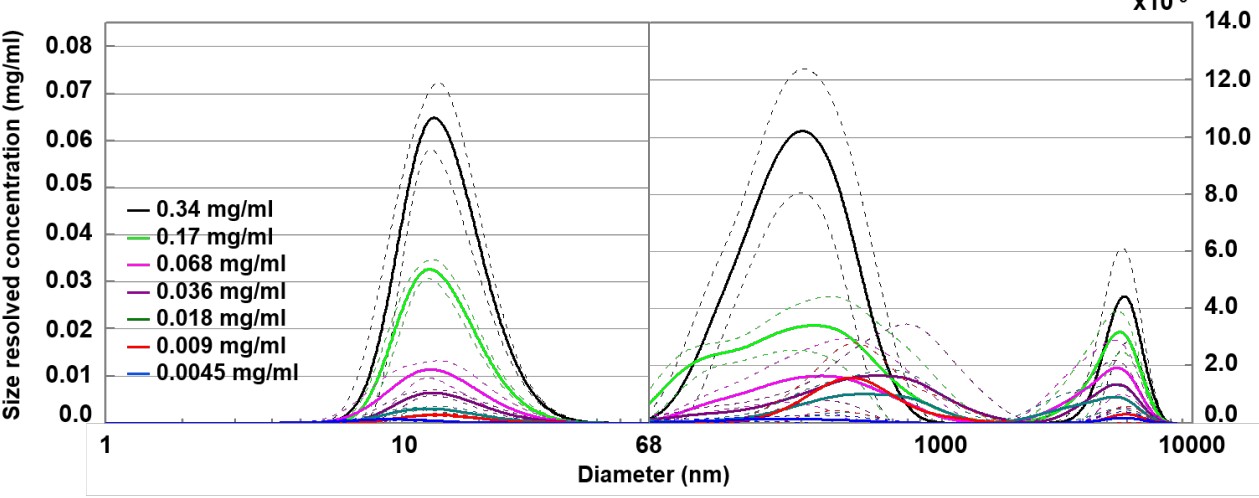

**Figure 7.** Hydrodynamic diameter of apoferritin (batch 2) depending on solution concentration measured with DLS with concentrations listed in the legend. Size resolved concentrations are obtained by multiplying the volume-weighted distribution with the apoferritin concentration. The main peak at 12.5 ± 0.3 nm corresponds to cage monomers and cage oligomers. The weak peaks above 68 nm (note the different y-axis scale) arise from cage aggregates. Solid lines are averages of multiple measurements, dashed lines indicate one standard deviation.

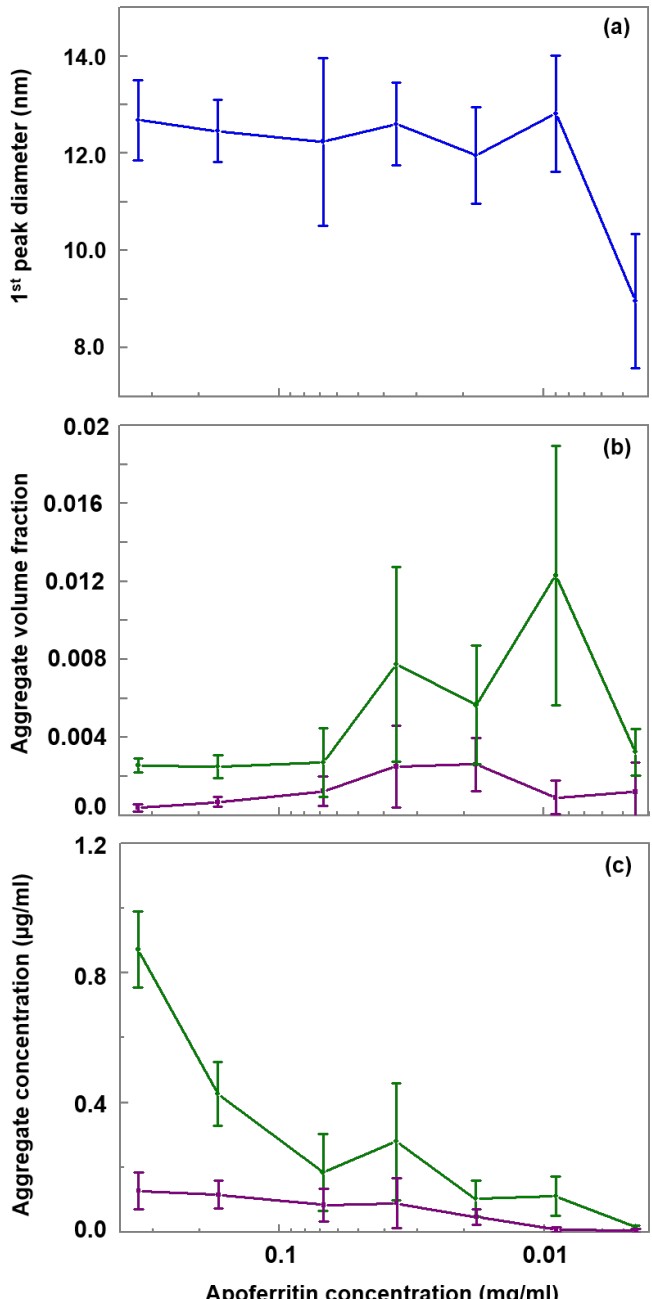

**Figure 8.** Evaluation of DLS measurements. (a) Hydrodynamic diameter of the main peak detected by DLS vs apoferritin solution concentration. (b) Relative abundance of aggregates (obtained by summing the volume-weighted distribution from 68 to $10^4$ nm). (c) Absolute concentration of aggregates in the solution (obtained by multiplying the aggregate volume-weighted distribution with the apoferritin solution concentration). Green lines in panels b and c represent aggregates from 68 to 1990 nm while purple lines represent aggregates from 2300 to $10^4$ nm. The vertical bars indicate one standard deviation.

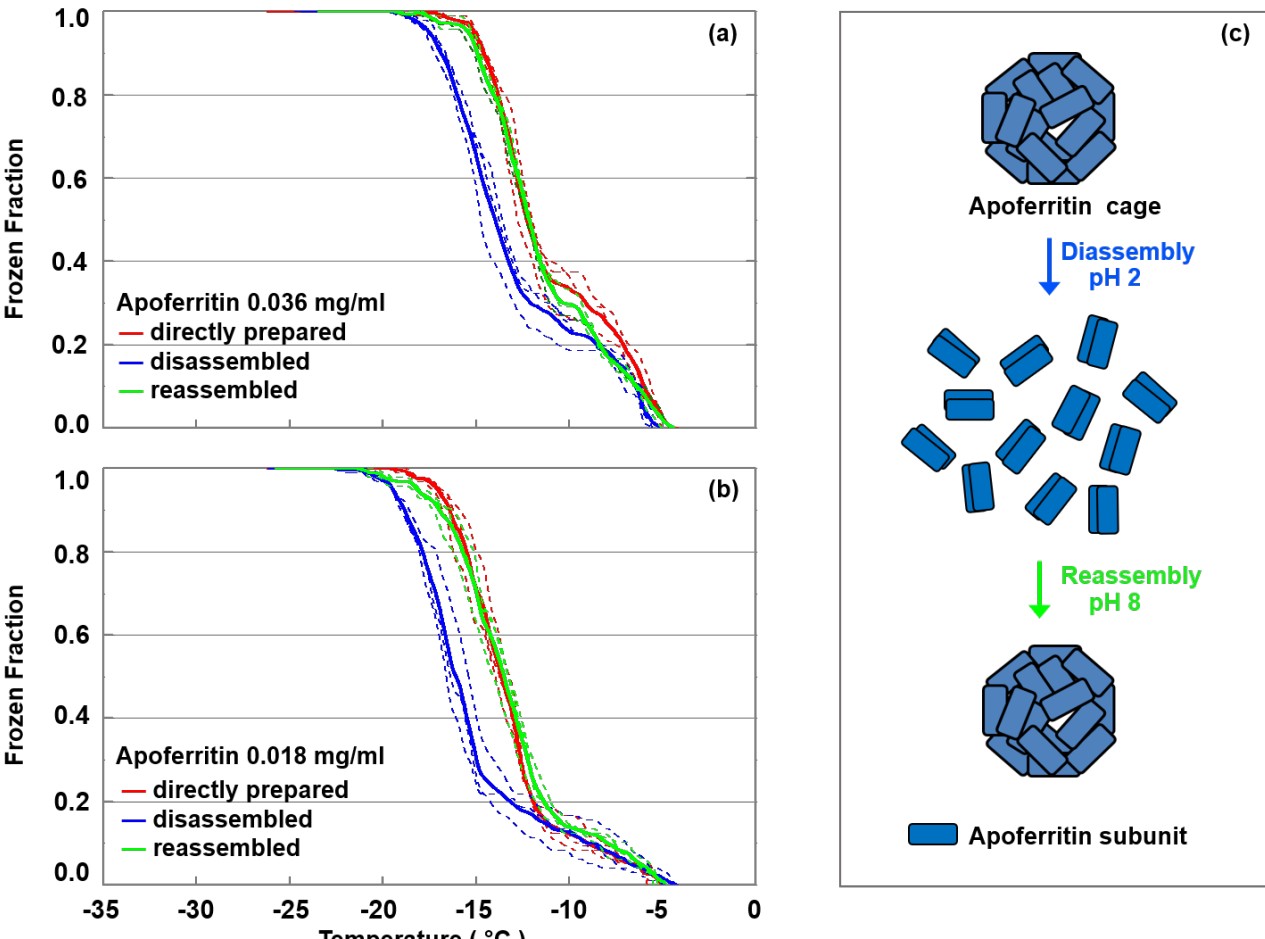

**Figure 9.** Disassembly and reassembly experiment performed with apoferritin (batch 2) 0.036 mg/ml (panel a) and 0.018 mg/ml (panel b). Frozen fractions are given for the directly prepared apoferritin, the disassembled apoferritin at pH 2 and the reassembled apoferritin at pH 8. (c) Schematic illustration of the disassembly and reassembly process: at pH 2 apoferritin undergoes disassembly into rod-like subunit dimers, increasing pH to 8 restores the fully assembled apoferritin cage.

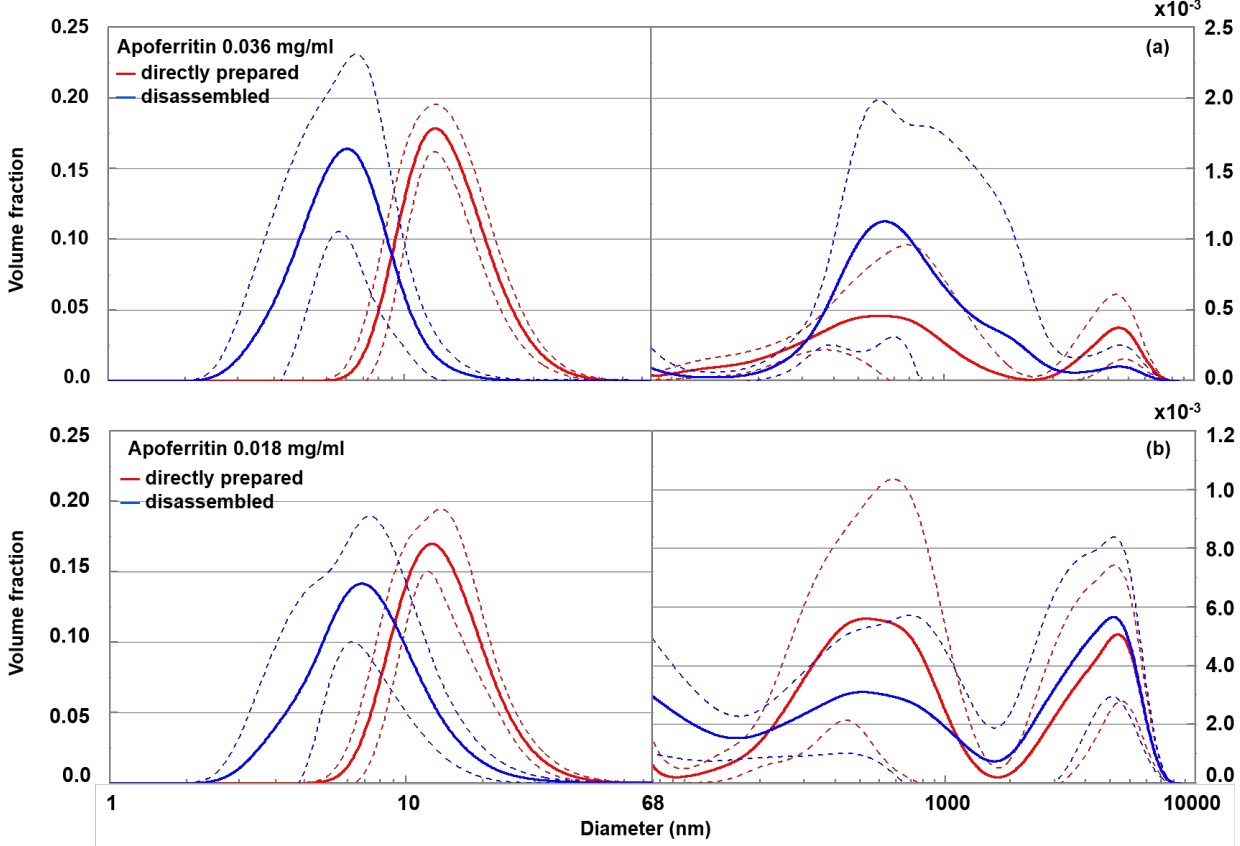

**Figure 10.** Volume-weighted distribution of directly prepared (red) and disassembled (blue) apoferritin, batch 2, with concentrations of 0.036 mg/ml (panel a) and 0.018 mg/ml (panel b). Solid lines are averages of multiple measurements, dashed lines indicate one standard deviation.

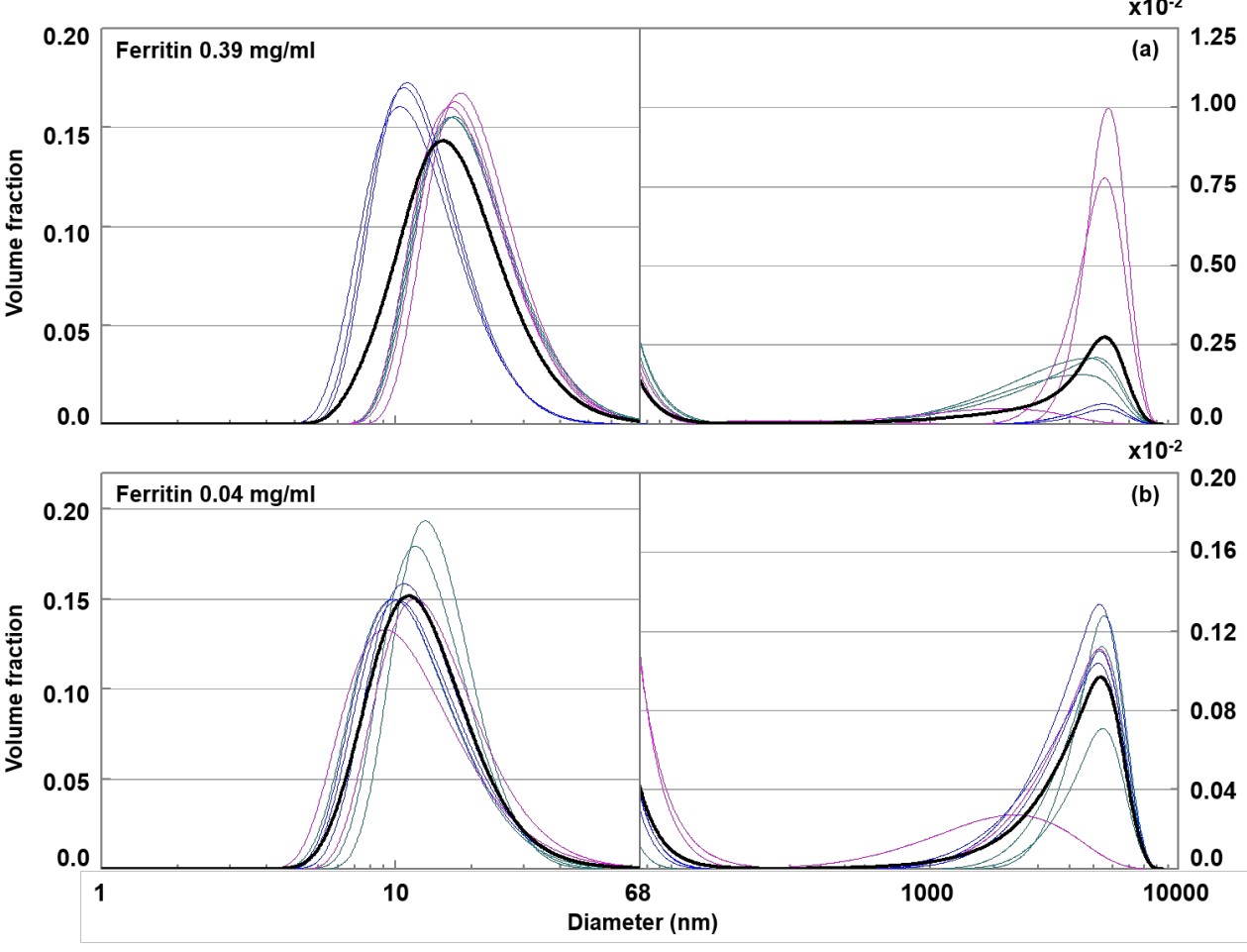

**Figure 11.** Volume-weighted distribution of ferritin with concentrations of 0.39 mg/ml (panel a) and 0.04 mg/ml (panel b). Green, pink and blue lines correspond to single measurements performed with three different preparations. The black line gives the average of all measurements.

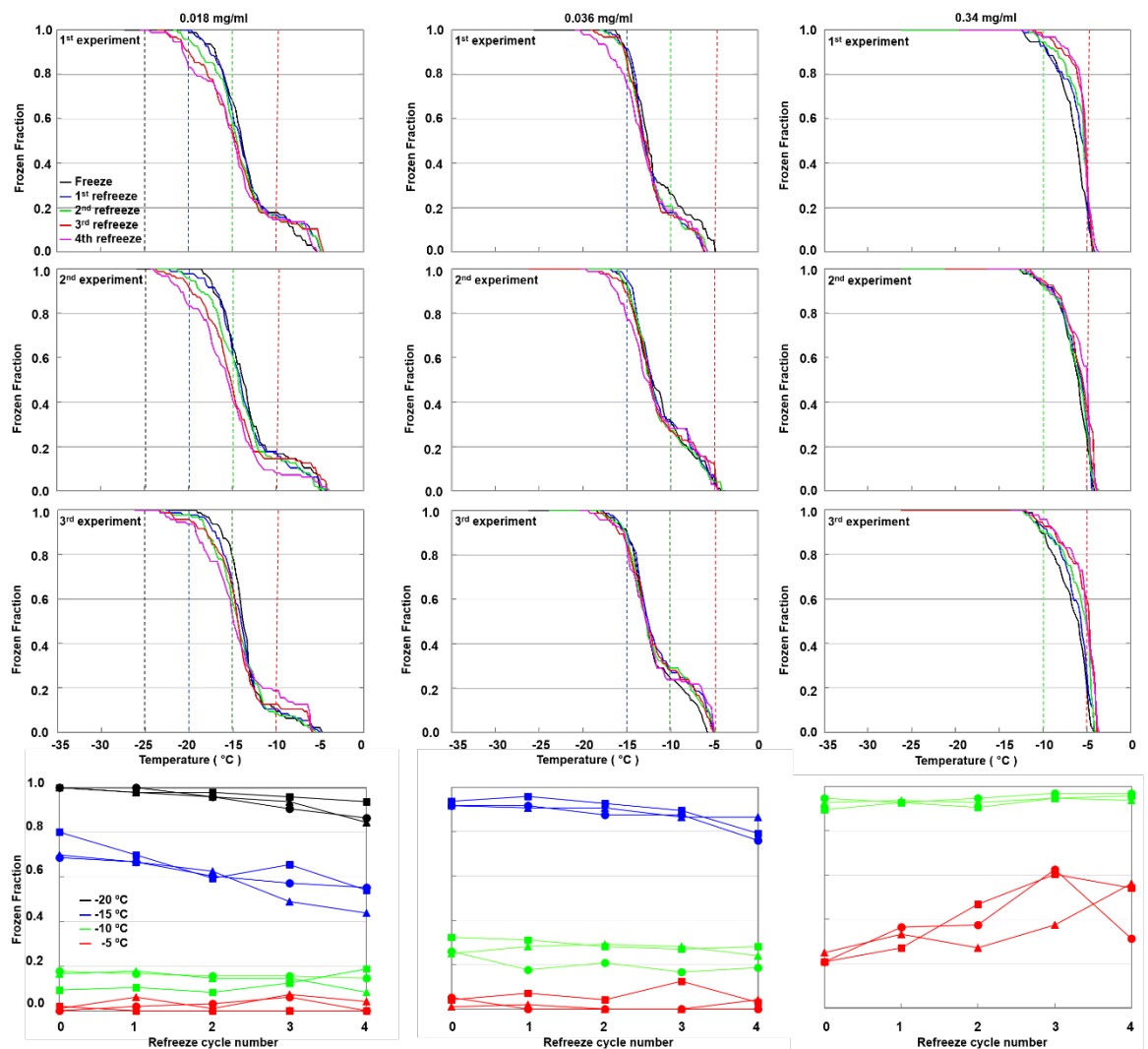

**Figure 12.** Refreeze experiments with apoferritin (batch 2) solutions with concentrations of 0.018 mg/ml (first column), 0.036 mg/ml (second column), and 0.34 mg/ml (third column). Frozen fractions are given for the first freeze-thaw cycle (cycle 0) and 4 refreeze cycles. For each concentration three independent preparations were investigated (rows 1 – 3). Experiments were analysed with respect to frozen fraction at -5, -10, -15, and -20°C (row 4).

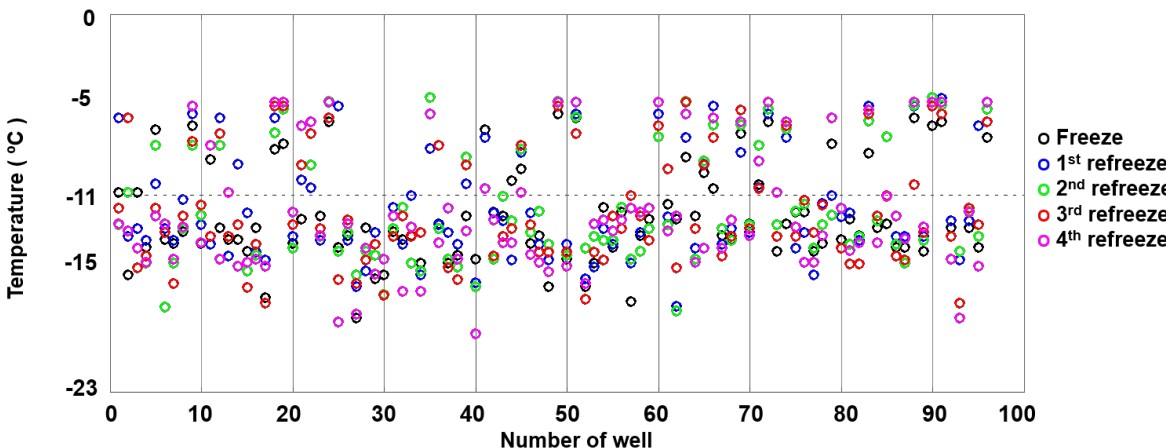

**Figure 13**. Well per well results of a refreeze experiment performed with apoferritin (batch 2), 2nd experiment with 0.036 mg/ml of Fig. 12.