# Peer review of "Protein aggregates nucleate ice: the example of apoferritin"

_Atmospheric Chemistry and Physics, 2019_

## Short Comment (SC1) · 4 Dec 2019

The statement in the introduction (page 2, line 39)" In screening experiments, fungi and lichen failed to show IN activity above -25°C, with the exception of Fusarium acuminatum and Fusarium avenaceum (Pouleur et al., 1992; Pummer et al., 2013)." is not supported by the literature since several studies found IN activity in other fungal species as well as lichen (e.g., Fröhlich-Nowoisky et al., 2015; Haga et al., 2013; 2014; Huffmann et al., 2013; Kieft 1988; Kieft and Ahmadjian, 1989; Kieft and Ruscetti, 1990; Moffet et al., 2015; Morris et al., 2013).

Moreover, the authors cite Fröhlich-Nowoisky et al., 2015 (page 3, line 1) to support their statement "Surveys of the IN ability of pollen showed that only few types were

active, the most active ones stemming from birch and conifer trees, yet, only at temperatures below -9°C".

This should be corrected as Fröhlich-Nowoisky et al., 2015 performed a screening of soil fungi and found ice nucleation activity in the widespread soil fungus Mortierella alpina.

References:

Fröhlich-Nowoisky, J., Hill, T. C. J., Pummer, B. G., Yordanova, P., Franc, G. D., and Pöschl, U.: Ice nucleation activity in the widespread soil fungus Mortierella alpina, Biogeosciences, 12, 1057–1071, doi:10.5194/bg-12-1057-2015, 2015.

Fröhlich-Nowoisky, J., Hill, T. C. J., Pummer, B. G., Yordanova, P., Franc, G. D., and Pöschl, U.: Ice nucleation activity in the widespread soil fungus Mortierella alpina, Biogeosciences, 12, 1057–1071, doi:10.5194/bg-12-1057-2015, 2015.

Haga, D.I., Iannone, R., Wheeler, M.J., Mason, R., Polishchuk, E.A., Fetch, T., van der Kamp, B.J., McKendry, I.G., Bertram, A.K.:Ice nucleation properties of rust and bunt fungal spores and their transport to high altitudes, where they can cause heterogeneous freezing. J. Geophys. Res. Atmos. 118, 7260–7272. http://dx.doi.org/10.1002/jgrd.50556, 2013.

Haga, D. I., Burrows, S. M., Iannone, R., Wheeler, M. J., Mason, R. H., Chen, J., Polishchuk, E. A., Pöschl, U., and Bertram, A. K.: Ice nucleation by fungal spores from the classes Agaricomycetes, Ustilaginomycetes, and Eurotiomycetes, and the effect on the at- mospheric transport of these spores, Atmos. Chem. Phys., 14, 8611–8630, doi:10.5194/acp-14-8611-2014, 2014.

Huffman, J. A., Prenni, A. J., DeMott, P. J., Pöhlker, C., Mason, R. H., Robinson, N. H., Fröhlich-Nowoisky, J., Tobo, Y., Després, V. R., Garcia, E., Gochis, D. J., Harris, E., Müller-Germann, I., Ruzene, C., Schmer, B., Sinha, B., Day, D. A., Andreae, M. O., Jimenez, J. L., Gallagher, M., Kreidenweis, S. M., Bertram, A. K., and Pöschl, U.:
High concentrations of biological aerosol particles and ice nuclei during and after rain, Atmos. Chem. Phys., 13, 6151–6164, doi:10.5194/acp13-6151-2013, 2013.

Kieft, T. L.: Ice nucleation activity in lichens, Appl. Environ. Microbiol., 54, 1678–1681, 1988.

Kieft, T. L. and Ahmadjian, V.: Biological ice nucleation activity in lichen mycobionts and photobionts, Lichenol., 21, 355–362, 1989.

Kieft, T. L. and Ruscetti, T.: Characterization of biological ice nuclei from a lichen, J. Bacteriol., 172, 3519–3523, 1990.

Moffett, B.F., Getti, G., Hill, T.C.J.: Ubiquity of ice nucleation in lichen – possible atmospheric implications. Lindbergia 38, 39–43, 2015.

Morris, C. E., D. C. Sands, C. Glaux, J. Samsatly, S. Asaad, A. R. Moukahel, F. L. T. Gonçalves, and E. K. Bigg: Urediospores of rust fungi are ice nucleation active at > −10 âŮęC and harbor ice nucleation active bacteria. Atmos. Chem. Phys. 13:4223-4233. http://www.atmos-chem-phys.net/13/4223/2013/acp-13-4223-2013, 2013.

---

## Referee Comment (RC1) · Anonymous Referee #1 · 17 Dec 2019

This manuscript describes observations of high temperature ice nucleation activity of some common proteins and a virus. The introduction and discussion point toward their relevance as potential cloud aerosols. There is, however, very little background on the properties of proteins/compounds that serve as ice nucleators. The efficacy of an aerosol to serve as an INP largely depends on its chemical/mineral makeup, morphology, and size. None of these properties are introduced and sufficiently discussed. In addition, the rationale for the experimental design and selection of these specific proteins for analysis is not presented.

I believe the results presented are interesting in that they provide insight into the (unexpected) types of proteinaceous compounds that can serve as efficient INPs. However, there are a number of items in the presentation that must be clarified before this article

should be considered for publication.

Specific comments:

Pg. 2, Lines 23-25: The first two sentences give the impression of a discussion on marine INPs but are unrelated to the content that follows in this paragraph. Please considering editing for clarity.

Pg. 2, Line 30 and throughout manuscript: Please do not capitalized the species name, i.e, Pseudomonas syringae.

Pg. 2 Line 33: Suggest replacing "populate surfaces" with "leaf surfaces" or describing the bacteria as epiphytes.

Pg. 3, Line 4: "Here we focus on proteins, which are the most or second-most important biological INPs." Please explain the basis for this statement. Presumably it is in reference to an atmospheric context.

Pg. 5, Line 35: Please explain what this means (skimming David et al. 2019 didn't help easily figure this out).

Pg. 6, Lines 30-31: This step would create water loss due to evaporation during the treatment. Was the protein concentration determined after the treatment or was concentration corrected for the evaporated volume?

Pg. 7, Lines 21-23: Please explain the rationale for the protein concentrations used in the experiments. For example, why were all not tested at the same molar concentration?

Pg. 8, Lines 12-13: Please explain the basis of this conclusion. Is there evidence from other measurements that the quaternary structure of the protein was disrupted in batch 2?

Pg. 8, Lines 19-21: Unclear how the presence/absence of Fe is inferred. Would dialysis be expected to remove Fe bound to the ferritin protein?

Pg. 13, Lines 13-14: First, the IN protein is in the outer membrane of these bacteria, not their cell membrane. Second, I'm not certain this comparison is a good one since heat-treated IN proteins of P. syringae lose their activity.

Pg. 14, Lines 13-14: The statement that cells must be disrupted to display IN activity is not accurate (e.g., Christner et al. 2008, 319:1214).

Pg. 14, Lines 22-23: I had trouble following this argument. Are you referring to aggregation in the wet phase? If not, how do dry aerosols become diluted?

Figure 2: What are the dotted lines co-plotted? Confidence intervals?

––––––––––––––––––––––––

---

## Referee Comment (RC2) · Anonymous Referee #2 · 31 Dec 2019

The authors describe the ice nucleation active entities of biological materials and highlight the ice nucleation activity of proteins and viruses. The bulk freezing experiment DRINCZ is used to investigate 96 wells at the same time. Common proteins were screened; a particular focus was on ferritin in its iron-containing and iron-free modification. The authors conclude that ice nucleation activity seems to be a common feature of diverse proteins, irrespective of their function, but arising only rarely, most probably through defective folding or aggregation to structures that are ice nucleation active

This paper is well-written and the topic fits into the journal Atmospheric Chemistry and Physics. The paper should be published after some changes, which are listed below:

1. Thoroughly describe the basic principles of proteinaceous ice nucleation in the introduction. How did other authors describe the correlation between the sizes of the

proteins/aggregates and their ice nucleation activity? What are the differences between free proteins and those embedded in the outer membrane? Quote Pummer et al. 2015 and literature quoted within.

2. When mentioning the ice nucleation activity of Pseudomonas syringae, you might also explain the aging of P. syringae, which drops the freezing temperature by more than 5°C only due to storage in the dark at temperatures below 0°C (see e.g. Häusler et al. 2018). What are the reasons for the aging effect? Changing of size can be excluded at these conditions. How does this effect correlate to your findings?

3. You might consider that aggregation is important not only between proteins but also between proteins and polysaccharides (e.g. cellulose). Please quote Felgitsch et al. 2018 and literature quoted within.

4. P. syringae has large ice-templating sites, which most other proteins do not exhibit. Aggregation and deective folding will not generate such ice-templating sites. What kind of ice nucleation mechanism do you anticipate for the proteins in your study?

References

T. Häusler, L. Witek, L. Felgitsch, R. Hitzenberger and H. Grothe, Freezing on a Chip—A New Approach to Determine Heterogeneous Ice Nucleation of Micrometer-Sized Water Droplets, Atmosphere, 9, 140; doi:10.3390/atmos9040140, 2018 L. Felgitsch, P. Baloh, J. Burkart, M. Mayr, M. E. Momken, T. M. Seifried, P. Winkler, D. G. Schmale III, and H. Grothe, Birch leaves and branches as a source of ice-nucleating macromolecules, Atmos. Chem. Phys., 18, 16063–16079, https://doi.org/10.5194/acp-18-16063-2018, 2018 B. G. Pummer, C. Budke, S. Augustin-Bauditz, D. Niedermeier, L. Felgitsch, C. J. Kampf, R. G. Huber, K. R. Liedl, T. Loerting, T. Moschen, M. Schauperl, M. Tollinger, C. E. Morris, H. Wex, H. Grothe, U. Pöschl, T. Koop, and J. Fröhlich-Nowoisky, Ice nucleation by water-soluble macromolecules, Atmos. Chem. Phys., 15, 4077–4091, https://doi.org/10.5194/acp-15-4077-2015, 2015

---

## Author Comment (AC1) · 31 Jan 2020

***Responses to short comment of Janine Fröhlich-Nowoisky***

*We thank Janine Fröhlich-Nowoisky for her corrections, which we implemented in the revised manuscript as detailed below (responses are in italic, text additions to the revised manuscript are in blue).*

The statement in the introduction (page 2, line 39) "In screening experiments, fungi and lichen failed to show IN activity above -25°C, with the exception of Fusarium acuminatum and Fusarium avenaceum (Pouleur et al., 1992; Pummer et al., 2013)." is not supported by the literature since several studies found IN activity in other fungal species as well as lichen (e.g., Fröhlich-Nowoisky et al., 2015; Haga et al., 2013; 2014; Huffmann et al., 2013; Kieft 1988; Kieft and Ahmadjian, 1989; Kieft and Ruscetti, 1990; Moffet et al., 2015; Morris et al., 2013).

*Thank you for pointing this out. We corrected the text and added the suggested references. The text reads now like this (page 2, line 38 – page 3, line 6 of the revised manuscript):*

Screening experiments revealed IN activity of lichen samples from a variety of locations with freezing onset temperatures up to -5°C (Moffett et al., 2015), and even up to -2.3°C (Kieft, 1988). The IN activity was found to originate primarily from the mycobiont (Kieft and Ahmadjian, 1989), providing evidence for a fungal rather than bacterial source of IN activity (Kieft and Ruscetti, 1990). The sites seem to be proteinaceous, although they are less sensitive to heat and pH variation compared with ice nucleating proteins expressed by *P. syringae* (Kieft and Ahmadjian, 1989; Kieft and Ruscetti, 1990; 1992). In screening experiments, most fungi failed to show IN activity above -20°C with few exceptions such as *Fusarium acuminatum* and *Fusarium avenaceum* (Pouleur et al., 1992; Pummer et al., 2013; Haga et al. 2013; 2014). Yet, IN active fungi with freezing onsets as high as -5°C could be identified in bioaerosols (Huffman et al., 2013) and in soils (Fröhlich-Nowoisky et al., 2015). Heat resistance and insensitivity to pH variation suggests that the IN active entity is more similar to the ones of lichen than to bacterial ones (Pouleur et al., 1992).

Moreover, the authors cite Fröhlich-Nowoisky et al., 2015 (page 3, line 1) to support their statement "Surveys of the IN ability of pollen showed that only few types were active, the most active ones stemming from birch and conifer trees, yet, only at temperatures below -9°C". This should be corrected as Fröhlich-Nowoisky et al., 2015 performed a screening of soil fungi and found ice nucleation activity in the widespread soil fungus Mortierella alpina.

*We removed the reference in this sentence and refer now to Fröhlich-Nowoisky et al. (2015) in the paragraph about fungi (page 3, line 5) in the revised manuscript.*

References:

Fröhlich-Nowoisky, J., Hill, T. C. J., Pummer, B. G., Yordanova, P., Franc, G. D., and Pöschl, U.: Ice nucleation activity in the widespread soil fungus Mortierella alpina, Biogeosciences, 12, 1057–1071, doi:10.5194/bg-12-1057-2015, 2015.

Fröhlich-Nowoisky, J., Hill, T. C. J., Pummer, B. G., Yordanova, P., Franc, G. D., and Pöschl, U.: Ice nucleation activity in the widespread soil fungus Mortierella alpina, Biogeosciences, 12, 1057–1071, doi:10.5194/bg-12-1057-2015, 2015.

Haga, D.I., Iannone, R., Wheeler, M.J., Mason, R., Polishchuk, E.A., Fetch, T., van der Kamp, B.J., McKendry, I.G., Bertram, A.K.:Ice nucleation properties of rust and bunt fungal spores and their transport to high altitudes, where they can cause heterogeneous freezing. J. Geophys. Res. Atmos. 118, 7260–7272. http://dx.doi.org/10.1002/ jgrd.50556, 2013.

Haga, D. I., Burrows, S. M., Iannone, R., Wheeler, M. J., Mason, R. H., Chen, J., Polishchuk, E. A., Pöschl, U., and Bertram, A. K.: Ice nucleation by fungal spores from the classes Agaricomycetes, Ustilaginomycetes, and Eurotiomycetes, and the effect on the at- mospheric transport of these spores, Atmos. Chem. Phys., 14, 8611–8630, doi:10.5194/acp-14-8611-2014, 2014.

Huffman, J. A., Prenni, A. J., DeMott, P. J., Pöhlker, C., Mason, R. H., Robinson, N. H., Fröhlich-Nowoisky, J., Tobo, Y., Després, V. R., Garcia, E., Gochis, D. J., Harris, E., Müller-Germann, I., Ruzene, C., Schmer, B., Sinha, B., Day, D. A., Andreae, M. O., Jimenez, J. L., Gallagher, M., Kreidenweis, S. M., Bertram, A. K., and Pöschl, U.: High concentrations of biological aerosol particles and ice nuclei during and after rain, Atmos. Chem. Phys., 13, 6151–6164, doi:10.5194/acp13-6151-2013, 2013.

Kieft, T. L.: Ice nucleation activity in lichens, Appl. Environ. Microbiol., 54, 1678–1681, 1988.

Kieft, T. L. and Ahmadjian, V.: Biological ice nucleation activity in lichen mycobionts and photobionts, Lichenol., 21, 355–362, 1989.

Kieft, T. L. and Ruscetti, T.: Characterization of biological ice nuclei from a lichen, J. Bacteriol., 172, 3519–3523, 1990.

Moffett, B.F., Getti, G., Hill, T.C.J.: Ubiquity of ice nucleation in lichen – possible atmospheric implications. Lindbergia 38, 39–43, 2015.

Morris, C. E., D. C. Sands, C. Glaux, J. Samsatly, S. Asaad, A. R. Moukahel, F. L. T. Gonçalves, and E. K. Bigg: Urediospores of rust fungi are ice nucleation active at > −10 â˚Ue¸C and harbor ice nucleation active bacteria. Atmos. Chem. Phys. 13:42234233. http://www.atmos-chem-phys.net/13/4223/2013/acp-13-4223-2013, 2013.

---

## Author Comment (AC2) · 31 Jan 2020

***Responses to Anonymous Referee #1***

*We thank the reviewer for his/her constructive comments that we address below line by line (responses are in italic, text additions to the revised manuscript are in blue).*

This manuscript describes observations of high temperature ice nucleation activity of some common proteins and a virus. The introduction and discussion point toward their relevance as potential cloud aerosols. There is, however, very little background on the properties of proteins/compounds that serve as ice nucleators. The efficacy of an aerosol to serve as an INP largely depends on its chemical/mineral makeup, morphology, and size. None of these properties are introduced and sufficiently discussed.

*We added a paragraph to the revised manuscript discussing surface properties that are considered to promote ice nucleation (Page 3, lines 13 – 25):*

Heterogeneous ice nucleation is considered to arise from the ability of surfaces to order water molecules in an ice-like pattern. The arrangement of water molecules at a surface depends on surface charge and functional groups (Glatz and Sarupria, 2016; Abdelmonem et al., 2017; Pummer et al., 2015). A relevant role is attributed to surface OH and NH groups that are able to form hydrogen bonds to water molecules. Their number and arrangement have been used to explain IN activity of different mineral surfaces (Pedevilla et al., 2007; Hu and Michaelides, 2007; Glatz and Sarupria, 2018; Kumar et al., 2019b). A lattice match between ice and the ice-nucleating agent is often considered a prerequisite for heterogeneous ice nucleation. Yet, while some IN active substances such as AgI (Marcolli et al., 2016) and 2D-crystalline films formed by long-chain alcohols (Popovitz-Biro et al., 1994; Zobrist et al., 2007; Qiu et al., 2017) exhibit a lattice match, others such as quartz (Kumar et al., 2019a) do not, and even others such as $BaF_2$ exhibit a lattice match but fail to be IN active (Conrad et al., 2005). The difficulty to pinpoint surface properties that are required for heterogeneous ice nucleation may be explained by growing evidence that it is not the whole surface that is able to nucleate ice but just special nucleation sites (Vali, 2014; Vali et al., 2015), which may arise through defects or impurities. Applying classical nucleation theory to heterogeneous ice nucleation yields nucleation site areas in the range of $10 - 50$ nm$^2$ required to host an ice embryo of critical size (Kaufmann et al., 2017).

In addition, the rationale for the experimental design and selection of these specific proteins for analysis is not presented.

*The selected proteins cover a broad variety of forms, sizes and functions as outlined in Sect. 2.1. The rationale was to elucidate whether proteinaceous material has an inherent ability to nucleate ice, irrespective of its function. We add a sentence explaining the rationale of this study (page 3, lines 31 – 32):*

So far, investigations have been focused on proteins that are expressed by organisms to nucleate ice. Here we examine whether proteins as a type of macromolecules have an inherent ability to nucleate ice.

I believe the results presented are interesting in that they provide insight into the (unexpected) types of proteinaceous compounds that can serve as efficient INPs. However, there are a number of items in the presentation that must be clarified before this article should be considered for publication.

Specific comments:

Pg. 2, Lines 23-25: The first two sentences give the impression of a discussion on marine INPs but are unrelated to the content that follows in this paragraph. Please considering editing for clarity.

*These three lines are a paragraph on their own. This paragraph and the previous one discuss sources of biological aerosols in terms of areas (terrestrial and marine). The next paragraph treats the nature of biological material that has been found to nucleate ice.*

Pg. 2, Line 30 and throughout manuscript: Please do not capitalized the species name, i.e, Pseudomonas syringae.

*Thanks for pointing this out. We have corrected to "syringae" throughout the manuscript.*

Pg. 2, Line 33: Suggest replacing "populate surfaces" with "leaf surfaces" or describing the bacteria as epiphytes.

*We specify "leaf surfaces" in the revised manuscript.*

Pg. 3, Line 4: "Here we focus on proteins, which are the most or second-most important biological INPs." Please explain the basis for this statement. Presumably it is in reference to an atmospheric context.

*In response to a comment of referee #2, we replaced this sentence by:*

So far, investigations have been focused on proteins that are expressed by organisms to nucleate ice. Here we examine whether proteins as a type of macromolecules have an inherent ability to nucleate ice.

Pg. 5, Line 35: Please explain what this means (skimming David et al. 2019 didn't help easily figure this out).

*We now mention the purpose of the bath leveler in the text:*

…the bath leveler, which keeps the ethanol bath level constant during a cooling ramp, was used as described in David et al. (2019).

Pg. 6, Lines 30-31: This step would create water loss due to evaporation during the treatment. Was the protein concentration determined after the treatment or was concentration corrected for the evaporated volume?

*For the heat treatments, the bottles were loosely closed by a cap so that water loss was low (less than 2 %). We did not correct this water loss, as it was low and we did not exactly quantify it. For clarification, we add to the revised manuscript:*

To prevent water loss, the bottles were loosely covered by a cap.

Pg. 7, Lines 21-23: Please explain the rationale for the protein concentrations used in the experiments. For example, why were all not tested at the same molar concentration?

*For heterogeneous ice nucleation, the surface area is considered relevant. When the surface area is not determined or not well defined, mass concentration is usually used. Plotting active site densities per mass is a common way to normalize IN activity of biological material (see e.g. Kanji et al., 2017, Fig. 1-5 or Pummer et al., 2015). Therefore, we used for all proteins the same mass concentration in the screening experiment. For the virus, we had to use a lower concentration as it was provided to us at this concentration.*

Pg. 8, Lines 12-13: Please explain the basis of this conclusion. Is there evidence from other measurements that the quaternary structure of the protein was disrupted in batch 2?

*In both batches, the main component is the correctly folded and assembled apoferritin or ferritin. Yet, there is a tiny fraction of misfolded apoferritin/ferritin that is most probably involved in the formation of larger aggregates. We show later in the manuscript that we were able to confirm the presence of aggregates by DLS.*

Pg. 8, Lines 19-21: Unclear how the presence/absence of Fe is inferred. Would dialysis be expected to remove Fe bound to the ferritin protein?

*Dialysis is not expected to remove Fe bound to the ferritin protein. We improve the sentence to make clear that we exclude an effect of Fe that is bound to ferritin:*

Overall, the IN activity of ferritin is lower than the one of apoferritin, which makes it unlikely that iron plays an active part in ice nucleation by ferritin.

Pg. 13, Lines 13-14: First, the IN protein is in the outer membrane of these bacteria, not their cell membrane. Second, I'm not certain this comparison is a good one since heat-treated IN proteins of P. syringae lose their activity.

*We refer only to the outermost membrane, which is now obvious in the text. In addition, please note that these lines do not refer to heat treatment.*

Pg. 14, Lines 13-14: The statement that cells must be disrupted to display IN activity is not accurate (e.g., Christner et al. 2008, 319:1214).

*We were thinking here of ferritin and other proteins that are not located on the outer membrane of organisms and will not be in contact with the environment when the organism is intact. However, the reviewer is correct that this does not need to be the case in general. Therefore, we removed this sentence in the revised manuscript.*

Pg. 14, Lines 22-23: I had trouble following this argument. Are you referring to aggregation in the wet phase? If not, how do dry aerosols become diluted?

*We are referring here to dilution during cloud droplet activation and concentration in the aerosol particles.*

Figure 2: What are the dotted lines co-plotted? Confidence intervals?

*In all frozen-fraction plots, the dotted lines are the results of individual freezing runs. In the revised manuscript, we have added this information now also to Fig. 2.*

---

## Author Comment (AC3) · 31 Jan 2020

***Responses to Anonymous Referee #2***

*We thank the reviewer for his/her constructive comments that we address below point by point (responses are in italic, text additions to the revised manuscript are in blue).*

The authors describe the ice nucleation active entities of biological materials and highlight the ice nucleation activity of proteins and viruses. The bulk freezing experiment DRINCZ is used to investigate 96 wells at the same time. Common proteins were screened; a particular focus was on ferritin in its iron-containing and iron-free modification. The authors conclude that ice nucleation activity seems to be a common feature of diverse proteins, irrespective of their function, but arising only rarely, most probably through defective folding or aggregation to structures that are ice nucleation active This paper is well-written and the topic fits into the journal Atmospheric Chemistry and Physics. The paper should be published after some changes, which are listed below:

1. Thoroughly describe the basic principles of proteinaceous ice nucleation in the introduction. How did other authors describe the correlation between the sizes of the proteins/aggregates and their ice nucleation activity? What are the differences between free proteins and those embedded in the outer membrane? Quote Pummer et al. 2015 and literature quoted within.

*In Sect. 3.3 (Comparison with other ice-nucleating proteins), we discuss aspects of proteinaceous ice nucleation and relate it with the IN activity of the proteins screened in this study. Moreover, in response to the request by the reviewer, we add a more general description of proteinaceous IN activity to the introduction (page 3, lines 26 – 34 of the revised manuscript):*

Taking surfaces that are large enough to host a critical ice embryo and have the ability to form hydrogen bonds to water molecules as requirements for IN activity, organic molecules with hydroxyl or carboxyl functionalities should potentially be able to induce freezing (Pummer et al., 2015). Indeed, microcrystalline cellulose has been found to nucleate ice up to -9°C (Hiranuma et al., 2015a). The IN activity from birch trees stems from macromolecules or aggregates of macromolecules which seem to involve polysaccharides (Pummer et al., 2012) and proteins (Tong et al., 2015; Felgitsch et al., 2018). Finally, ice-nucleating proteins expressed by *Pseudomonas* exhibit a repetition unit containing threonine amino acids with hydroxyl functional groups that are able to template ice. Aggregates involving only few of these proteins are water soluble and induce ice nucleation up to -7°C. Larger aggregates nucleate ice up to -2°C but require the intact outer cell membrane to be stable (Polen et al., 2016; Zachariassen and Kristiansen, 2000).

2. When mentioning the ice nucleation activity of Pseudomonas syringae, you might also explain the aging of P. syringae, which drops the freezing temperature by more than 5°C only due to storage in the dark at temperatures below 0°C (see e.g. Häusler et al. 2018). What are the reasons for the aging effect? Changing of size can be excluded at these conditions. How does this effect correlate to your findings?

*A decrease of IN activity with sample storage time also occurs in some mineral dusts. In the case of quartz particles, we recently showed that nucleation sites generated by milling disappear with time (Kumar et al., 2019). Thus, chemically reactive sites may be relevant for ice nucleation. In case of biological nucleation sites, the presence of chemically reactive sites is less likely. Alternatively, sites available for hydrogen bonding to water molecules may disappear with time through intramolecular bonding within the protein. Also, alteration of hydrogen bonding patterns, may influence aggregation. In the revised manuscript, we discuss this point by adding this sentence to the discussion on page 14, lines 28 – 31:*

Interestingly, the IN activity of Snomax® decreases with storage time indicating that the most efficient nucleation sites of Snomax® degrade with time (Polen et al., 2016; Häusler et al., 2018). This may be due to loss of free hydrogen bonding sites or disintegration of larger aggregates.

3. You might consider that aggregation is important not only between proteins but also between proteins and polysaccharides (e.g. cellulose). Please quote Felgitsch et al. 2018 and literature quoted within.

*This is a good point. In the revised manuscript, we quote Felgitsch et al. (2018) on page 3, line 10 of the revised manuscript:*

Moreover, IN activity has also been found in aqueous extracts of birch leaves and branches (Felgitsch et al., 2018).

*Furthermore, we discuss the IN activity of macromolecules from birch trees on page 3, lines 29 – 31 of the revised manuscript:*

The IN activity of birch tree extracts stems from macromolecules or aggregates of macromolecules which involve polysaccharides (Pummer et al., 2012) and proteins (Tong et al., 2015; Felgitsch et al., 2018) that may coaggregate.

4. P. syringae has large ice-templating sites, which most other proteins do not exhibit. Aggregation and defective folding will not generate such ice-templating sites. What kind of ice nucleation mechanism do you anticipate for the proteins in your study?

*This is a very good question, which we address now in more detail in Sect. 3.3 on page 15 (lines 6 – 12).*

Since the screened proteins all have characteristic freezing onset temperatures, their nucleation sites do not seem to be totally random but related to the protein structure. A templating effect may result from the pattern of hydrophilic and hydrophobic regions on alpha helices and beta sheets together with sites for hydrogen bonding responsible for the tertiary and quaternary structure. In misfolded proteins, these may be available to bind water molecules. Attached to ferritin are water molecules in inter-subunit interfaces through hydrogen bonds (Hempstead et al., 1997), which may be a starting point for ice embryos. Also, the outer protein shell features iron bonding sites (Massover, 1993), which may play a role in ice nucleation.

References

T. Häusler, L. Witek, L. Felgitsch, R. Hitzenberger and H. Grothe, Freezing on a Chipˇ AˇTA New Approach to Determine Heterogeneous Ice Nucleation of MicrometerSized Water Droplets, Atmosphere, 9, 140; doi:10.3390/atmos9040140, 2018

L. Felgitsch, P. Baloh, J. Burkart, M. Mayr, M. E. Momken, T. M. Seifried, P. Winkler, D. G. Schmale III, and H. Grothe, Birch leaves and branches as a source of ice-nucleating macromolecules, Atmos. Chem. Phys., 18, 16063–16079, https://doi.org/10.5194/acp18-16063-2018, 2018

B. G. Pummer, C. Budke, S. Augustin-Bauditz, D. Niedermeier, L. Felgitsch, C. J. Kampf, R. G. Huber, K. R. Liedl, T. Loerting, T. Moschen, M. Schauperl, M. Tollinger, C. E. Morris, H. Wex, H. Grothe, U. Pöschl, T. Koop, and J. FröhlichNowoisky, Ice nucleation by water-soluble macromolecules, Atmos. Chem. Phys., 15, 4077–4091, https://doi.org/10.5194/acp-15-4077-2015, 2015

*Kumar, A., Marcolli, C., and Peter, T.: Ice nucleation activity of silicates and aluminosilicates in pure water and aqueous solutions – Part 2: Quartz and amorphous silica, Atmos. Chem. Phys., 19, 6035–6058, doi:10.5194/acp-19-6035-2019, 2019.*

---

## Author Response (AR2)

Comments to the Author:
Dear Authors,

After careful evaluating the reviewers' comments and your responses, I have a few additional minor
comments. I would like to ask you to respond to these comments before proceeding to the publication of your
manuscript.

With kindest regards,

Daniel Knopf

*Dear Daniel*

*We thank you for carefully reading the manuscript and your comments, which we address below (responses are in italic).*

*Best regards*
*Claudia*

Minor comments:
P. 2, Line 12-14: I recommend to re-evaluate the definition of biological vs biogenic particles/INPs. Biological particles are bacteria, fungal spores, pollen, diatoms, etc. (Fröhlich-Nowoisky et al., 2016;Despres et al., 2012). The list you give, is a listing of biogenic particles, i.e. from biological entities derived.
Also, if you keep this list, you could cite (Hiranuma et al., 2015) for cellulose and our studies (Rigg et al., 2013;Wang et al., 2011) for humid material acting as INPs.

*Thank you for pointing this out. We replaced "biological" by "biogenic" and added the suggested papers.*

P. 2, Line 22-24: I think it would be fair to add references to the role of the marine environment as source of INPs. Like (Schnell, 1975); one of our 2011 papers, maybe (Alpert et al., 2011) and the studies by (Ladino et al., 2016) and (Wilson et al., 2015).

*We added the suggested references and the sentence: "Marine phytoplankton has been found IN active, both intact cells and exudates (Schnell, 1975; Alpert et al., 2011; Wilson et al., 2015; Ladino et al., 2016)."*

Just a comment: If you want to emphasize the importance of polysaccharides (like on p. 3, line 24), in the studies by (Ladino et al., 2016;Wilson et al., 2015), the exudate material that acted as INP is polysaccharidic. (Aller et al., 2017) has shown the presence of polysaccharides and proteinaceous material in marine aerosol.

*This is a good point. We add the following sentence to the discussion of IN activity of marine aerosol: "Similarly, the exudate material acting as INP in marine aerosol (Wilson et al., 2015; Ladino et al., 2016) was found to contain polysaccharidic and proteinaceous compounds (Aller et al., 2017)."*

P. 2, Line 25: I believe, those would qualify as biological INPs. Biological material cannot act as biogenic particle. It is either biological or biogenic (=biologically derived).

*We revised to: "Biological INPs include…"*

I struggled to understand some of the interpretation of the refreeze experiments, described on p. 13, line 6-22:

You present a couple of parameters in Tables 2 and 3, however, most of these parameters are not discussed in text or supplement. It is also not entirely clear how some of the numbers are derived. I struggled to understand the difference of the derivation of FF "over all wells and all cycles" vs. "evaluation of well by well"? A few more words on this would be helpful. If it distracts the main text, it could go in the supplement.

*We extended the text to make the parameters clearer. The explanation of FF over all wells and all cycles reads now: "For the three refreeze experiments performed with 0.036 mg/ml apoferritin samples, FF at -11°C (averaged over all 96 wells and the five cycles) was 0.219 for the 1st experiment (i.e. 29 wells were frozen at -11°C in the first cycle, 19 in the second, 20 in the third, 17 in the fourth, and 20 in the fifth cycle, yielding (29 + 19 + 20 + 17 + 20)/(5 x 96) = 0.219), 0.304 for the 2nd experiment, and 0.319 for the 3rd experiment."*

*and*

> *"However, evaluation of the well by well results (shown in Figs. 13 and S5) yielded fractions of wells always freezing at T > -11℃ from 0.156 (i.e. in the first experiment, 15 out of the 96 wells froze above -11℃ during all five cycles)  to 0.271 (see Table 2),…"*

However, my biggest struggle is the interpretation using FF as a kind of probability, where you take FF to the exponent of 4, trying to get the probability of for 4 trials/experiments. Like rolling the dice 4 times. Maybe I misunderstand something, but the FF value is not a probability. FF is a value of an accumulated distribution. If it would be a probability, then integration of the FF curve would result in 1, like for alpha-PDF etc. I try to come up with an example. If you roll a dice 10 times and normalize by, e.g., the maximum points (=60), you can get a "point fraction" as a function of throws. However, this is not a probability with which the result can be deduced. For each roll of the dice the probability for a given face is 1/6. Similarly, as you cooled the wells, each well has a chance to be frozen or liquid, i.e., 50%. So, I strongly doubt that you can use the FF value as a kind of probability and place the number of trials in the exponent, resulting in these very low (unrealistic) overall probabilities. Since I do not know how "well by well evaluation" is performed (what is the change in population/normalization numbers etc. compared to other method), for now it looks that there is overlap in the FF values from both methods. Frankly, speaking, it has to be like this, or? If not your data/figures would look rather different? This may change your final statement in this section. However, I find this a very valuable analysis. Of course, this entire discussion will be affected by uncertainties in FF but this is difficult to assess and likely beyond the scope of this study.

*For this calculation, we focused on FF at -11℃. Hence, there are two categories of wells: the ones frozen at -11℃ and the ones freezing at a lower temperature. Thus, it is like tossing coins (frozen at -11℃ is head and freezing lower is tail) with the difference that the probabilities of having head or tail are not given but need to be derived from the result of the experiment. Having an experiment with the 96-well plate with 5 freezing cycles is like tossing coin 96 x 5 = 480 times. In the first experiment with 0.036 mg/ml, it was 29 times head in the first cycle, followed by 19 in the second, then 20, 17, and 20 in cycles 3 to 5. From this, the probability of having head can be derived as: (29 + 19 + 20 + 17 + 20)/(5 x 96) = 0.219. (Probabilities need to be derived this way, when there is no analytical way to calculate them, for example mortality rates are calculated based on the cases that have been observed). Now, we focus only on the 29 wells with head in the first cycle and check whether they also showed head in the second cycle. The probability of this is 0.219. The probability of having head in the second and third cycle is 0.219 x 0.219, and the probability of having head in cycles 2 to 5 is 0.219 x 0.219 x 0.219 x 0.219. Thus, the probability that a well with head in cycle 1 also has head in cycles 2 to 5 is $0.219^4 = 0.0023$.*

Technical correction:
P. 9, line 16: Missing space "…SI). Apoferritin…"

*Corrected.*

[revised manuscript text omitted]